

# Atmospheric fate of a series of Methyl Saturated Alcohols (MSA): Kinetic and Mechanistic study

Inmaculada Colmenar[1,2], Pilar Martin[1,2], Beatriz Cabañas[1,2], Sagrario Salgado[1,2], Araceli Tapia[1,2], Inmaculada Aranda[1,2]

[1]Universidad de Castilla La Mancha, Departamento de Química Física, Facultad de Ciencias y Tecnologías Químicas, Avda. Camilo José Cela S/N, 13071 Ciudad Real, Spain

[2]Universidad de Castilla La Mancha, Instituto de Combustión y Contaminación Atmosférica (ICCA), Camino Moledores S/N, 13071 Ciudad Real, Spain

*Correspondence to*: Pilar Martín (mariapilar.martin@uclm.es)

**Abstract.** The atmospheric fate of a series of Methyl Saturated Alcohols (MSA) has been evaluated through the kinetic and reaction product studies with the main atmospheric oxidants. Rate coefficients (in $cm^3$ $molecule^{-1}$ $s^{-1}$ unit) measured at ~298K and atmospheric pressure (~ 740 Torr) were as follows: $(3.71 \pm 0.53) \times 10^{-10}$, $(1.91 \pm 0.65) \times 10^{-11}$ and $(2.92 \pm 1.38) \times 10^{-15}$ for reaction of E-4-methyl-cyclohexanol with Cl, OH and $NO_3$, respectively. $(2.70 \pm 0.55) \times 10^{-10}$ and $(5.57 \pm 0.66) \times 10^{-12}$ for reaction of 3,3-dimethyl-1-butanol with Cl and OH radical respectively and $(1.21 \pm 0.37) \times 10^{-10}$ and $(10.51 \pm 0.81) \times 10^{-12}$ for reaction of 3,3-dimethyl-2-butanol with Cl and OH radical respectively. The main detected products were 4-methylcyclohexanone, 3,3-dimethylbutanal and 3,3-dimethyl-2-butanone for the reactions of E-4-methyl-cyclohexanol, 3,3-dimethyl-1-butanol and 3,3-dimethyl-2-butanol respectively with the three oxidants. A tentative estimation of yields have been done obtaining the following ranges (25-60) % for 4-methylcyclohexanone, (40-60) % for 3,3-dimethylbutanal and (40-80) % for 3,3-dimethyl-2-butanone. Other products as HCOH, 2,2-dimethylpropanal and acetone have been identified in the reaction of 3,3-dimethyl-1-butanol and 3,3-dimethyl-2-butanol. The yields of these products indicate a hydrogen abstraction mechanism at different sites of the alkyl chain in the case of Cl reaction and a predominant site in the case of OH and $NO_3$ reactions, supported by SAR methods prediction.

Tropospheric lifetimes (τ) of these MSA have been calculated using the experimental rate coefficients. Lifetimes are in the range of 0.6-2 day**s** for OH reactions, 8-13 days for $NO_3$ radical reactions and 1-3 months for Cl atoms. In coastal areas the lifetime due to the reaction with Cl decreases to hours. The global tropospheric lifetimes calculated, and the polyfunctional compounds detected as reaction products in this work, imply that the Methyl Saturated Alcohols could contribute to ozone and nitrated compound formation at local, but also regional and even to global scale. Therefore, the use of large saturated alcohols as additives in biofuels must be taken with caution.



## 1. Introduction

Multitude of scientific studies about combustion emissions confirms that fossil fuels, especially diesel fuel, are the main responsible for air pollution. The loss of air quality and its consequences on health as well as global warming are some of the most important problems caused by air pollution (www.iea.org). These consequences have led governments to set restrictive limits for the presence of certain pollutants in the atmosphere, such is the case of particulate matter (PM) (EURO 6). This has led to develop biofuels (Sikarwara et al., 2017) as alternative to conventional ones.

Biodiesel are obtained from a transesterification process of animal or vegetable oils origin. Also, the fermentation of vegetal biomass gives methanol and / or ethanol (bioethanol). These lower alcohols have been used as fuels showing advantages as the reduction of the smoke, due to the presence of OH group that increases the oxygen content during the combustion process (Ren et al., 2008; Lapuerta et al., 2010; Sarathy et al., 2014; Sikarwara et al., 2017). However several studies have shown certain complications in the use of lower alcohols due to their low cetane number, high latent heat of vaporization and high resistance to auto-ignition (Karabektas and Hosoz, 2009). In order to avoid or to minimize these limitations, alcohol–diesel blends and alcohol–diesel emulsions have been used in diesel engines (Ozsezen et al., 2011). Other alternative is the use of longer alcohols (propanol, n-butanol, isobutanol and n-pentanol) with superior fuel properties than lower alcohols mixed with diesel fuel (Cheung, et al. 2014; Kumar and Saravanan, 2016).

The fact that the use of high alcohols is a good alternative to conventional fuels could support an important presence of these alcohols in the atmosphere. Therefore, previously to the massive use, it is necessary to study the reactivity of the large alcohols in atmospheric conditions, in order to establish and to evaluate their atmospheric impact.

Alcohols are present in the atmosphere from a wide variety of anthropogenic and biogenic sources (Calvert et al., 2011). Methanol, ethanol and isopropanol are some of the main alcohols detected in urban areas such as Osaka and Sao Paulo cities (Nguyen et al., 2001) with concentrations between 5.8-8.2 ppbv and 34.1-176.3 ppbv respectively. Others alcohols such as E-4-methylcyclohexanol, have been identified in the exhaust gas emissions of burning fuel blends containing 7 % v/v (B7) and 20 % v/v (B20) of soy bean/palm biodiesel (84 % / 16 %) (Lopes et al., 2014). 3,3-dimethyl-1-butanol is a glass forming material, used as a chemical intermediate in organic syntheses (www.capotchem.com). 3,3-dimethyl-2-butanol is a potential precursor for prohibited chemical weapons such as soman, a nerve agent (Murty et al., 2010). It is also used in conversion of ribose-and glucose-binding proteins into receptors for pinacolyl methyl phosphonic acid (Allert et al. 2004).

In the case of smaller alcohols, the knowledge of its reactivity is well established indicating that the degradation mechanism of saturated alcohols is mainly initiated by the H-abstraction from C-H bond. The H-abstraction from the OH group seems to be less favored (Grosjean et al., 1997). According to literature (Atkinson and Arey, 2003; Atkinson et al., 2006; Calvert et al., 2011; Caravan et al., 2015; Mellouki et al, 2015), the main degradation route of saturated alcohols in the atmosphere is the reaction with OH radicals during day time. Kinetics with chlorine atoms are expected to be high, therefore reactions with Cl could also be an important degradation route, especially in coastal areas where concentration peaks of chlorine can be found. Reactions with ozone ($k \leq 10^{-20}$ cm$^3$ molecule$^{-1}$ s$^{-1}$) and nitrate radical ($\sim 10^{-15}$ cm$^3$ molecule$^{-1}$ s$^{-1}$) are too low to have a significant contribution to its degradation. However, the determination of the rate coefficients and the reaction products of alcohols with the nitrate radical





are also necessary to better understand the general reactivity of alcohols in the atmosphere since the reactions with
this radical are a source of OH during the night-time (Finlayson-Pitts and Pitts, 2000).
Although in the last years some studies about reactivity of large alcohols have been made (Andersen et al. 2010;
Ballesteros et al., 2007; Calvert et al, 2011; Hurley et al., 2009; Moreno et al., 2012, 2014, Mellouki et al, 2015)
the kinetic and mechanistic database is still scarce. In the case of the 3,3-dimethyl-1-butanol and 3,3-dimethyl-2-
butanol (derives from 1-butanol, 2-butanol) there is a lack of information regarding the diurnal reactivity (Moreno
et al., 2014; Mellouki et al., 2015). Regarding to cyclic alcohols, only data concerning the reactivity of chlorine
atoms and OH radicals for cyclohexanol (Bradley et al., 2001; Ceacero-Vega et al., 2012) and OH for
cyclopentanol (Wallington et al., 1988) have been reported.
Therefore, in the present work, the study of gas phase reactions of some Methyl Saturated Alcohols (MSA): E-4-
methylcyclohexanol (4MCHexOH), 3,3-dimethyl-1-butanol (3,3DM1ButOH) and 3,3-dimethyl-2-butanol
(3,3DM2ButOH) with the main atmospheric oxidants has been done in order to complete the kinetic and
mechanism database, to improve our knowledge of the atmospheric chemistry of alcohols in special saturated
alcohols, and to assess a chemical's environmental impact.
**2. Experimental Section**
**2.1 Kinetic experiments**
The reactions of a series of Methyl Saturated Alcohols (MSA) with the main atmospheric oxidants have been
studied:

$$
\begin{array}{ll}
+\ \text{Cl} \longrightarrow \text{Products} & \text{(R1)} \\
+\ \text{OH} \longrightarrow \text{Products} & \text{(R2)} \\
+\ \text{NO}_3 \longrightarrow \text{Products} & \text{(R3)} \\
\end{array}
$$

$$
\begin{array}{ll}
+\ \text{Cl} \longrightarrow \text{Products} & \text{(R4)} \\
+\ \text{OH} \longrightarrow \text{Products} & \text{(R5)} \\
\end{array}
$$

$$
\begin{array}{ll}
+\ \text{Cl} \longrightarrow \text{Products} & \text{(R6)} \\
+\ \text{OH} \longrightarrow \text{Products} & \text{(R7)} \\
\end{array}
$$

Rate coefficients were determined using a relative rate method. This method relies on the assumption that the
organic compound (MSA) and the reference compound (R), are removed solely by their reactions with the oxidants
(Ox: OH, NO$_3$ radicals and Cl atoms):
Ox + MSA → Products (k$_S$)                                    (R8)
Ox + R → Products (k$_R$)                                      (R9)
On the assumption that the MSA and the reference compound are only consumed by reaction with the oxidant, the
kinetic treatment for the reactions expressed by R8 and R9 yields the following relationship;



$$ln\left(\frac{[MSA]_0}{[MSA]_t}\right) = \frac{k_{MSA}}{k_R} ln\left(\frac{[R]_0}{[R]_t}\right) \qquad (1)$$
where $[MSA]_0$, $[R]_0$, $[MSA]_t$, and $[R]_t$ are the initial concentrations and those at time $t$ for the Methyl Saturated
Alcohol and the Reference compound, respectively. Two reference compounds with each oxidant were used to
assure that the reference compound does not have any influence on overall rate coefficient.
According to Eq (I), a plot of ($\{ln([MSA]_0/[MSA]_t)\}$ versus $\{ln([R]_0/[R]_t)\}$) should be a straight line that passes
through the origin. The slope of this plot gives the ratio of rate coefficients $k_{MSA}/k_R$. The value of $k_{MSA}$ can therefore
be obtained if the rate coefficient $k_R$ of the reference compound is known.
Kinetic measurements were performed at room temperature (~298 K) and atmospheric pressure ~ (720 Torr) by
employing two experimental set-ups: All kinetic experiments, for the Cl and OH reactions, were performed using
a 50 L Pyrex® glass reaction chamber with a White cell that allowed a long pathlength up to 200 m coupled to a
FTIR spectrometer (Thermo Nicolet 5700) equipped with a KBr beam splitter and liquid nitrogen-cooled MCT
detector, as a detection technique. Typically, for each spectrum, 60 interferograms were co-added over 98 s and
approximately 30-40 spectra were recorded per kinetic experiment with a spectral resolution of 1 cm$^{-1}$. A spectral
subtraction procedure was used to derive the concentrations of reactant and reference compounds at time $t$=0 and
time $t$. Chlorine atoms were obtained by photolysis of $Cl_2$ at a wavelength of 360 nm using 8 actinic lamps. OH
radicals were produced by photolysis of Methyl nitrite, $CH_3ONO$, in the presence of NO in air. $CH_3ONO$ was
synthesized in the laboratory as described elsewhere (Taylor et al., 1980).
The reaction of $NO_3$ with 4MCHexOH was studied using a bigger reactor, a 150 L or 500 L Teflon® in order to
minimize the wall deposition and dilution effects of the consecutive additions of $N_2O_5$. Solid Phase Micro
Extraction fiber (SPME) as a pre-concentration sample method, followed by analysis on a Gas Chromatography-
Mass Spectrometry system with a Time of Flight analyzer (SPME/GC-TOFMS) (AccuTOF GCv, Jeol) was used.
Samples were collected by exposing a 50/30 mm DVB/CAR/PDMS Solid Phase Micro Extraction fiber (SPME,
SUPELCO) for 5 min during the reaction and then thermally desorbed for 15 min at 250 °C in the heated GC
injection port. A capillary column (30 m × 0.3 mm id × 1.0 mm film thickness, Tracsil TRB-1701, Teknokroma)
was used to separate the compounds. The chromatographic conditions used for the analysis were as follows:
injector, 250 °C; interface, 250 °C; oven initial temperature, 40 °C for 4 min; ramp, 30 °C min$^{-1}$ to 120 °C, held for
6 min, second ramp, 30 °C min$^{-1}$ to 200 °C, held for 3 min. Nitrate radicals were generated in situ in the dark by
the thermal decomposition of $N_2O_5$ (Atkinson et al., 1984, 1988). $N_2O_5$ was obtained mixing $O_3$ with excess of
$NO_2$ (Scott and Davidson, 1958). Previously to the kinetic experiments a series of tests in dark and photolysis
conditions were carried out to evaluate secondary reactions such as wall depositions and photo degradation
processes of reactants.
Range concentrations of reactants employed were as follows: 2-16 ppm of 4MCHexOH, 3-9 ppm of
3,3DM1ButOH and 3,3DM2ButOH, 9-30 ppm of $Cl_2$, 4-13 ppm of 1-butene and 2-methylpropene, 7-14 ppm of
propene and cyclohexene, 5-7 ppm of isopropanol, 3-5 of 2-methyl-2-butanol, 26-55 ppm of $CH_3ONO$, 20-60 ppm
of NO, 3-4 ppm of 2-ethyl-1-hexanol, 4-5 ppm of 1-butanol. For reactions of 4MCHexOH with nitrate radicals a
number of 2-5 additions of $N_2O_5$ with concentrations between 8-36 ppm were made per each experiment. $N_2$ and
synthetic air were used as bath gases for Cl, $NO_3$ and OH reactions, respectively.
**2.2 Product experiments**





The product study was carried out at room temperature (~298 K) employing the two detection set-ups mentioned
above, FTIR at a pressure of (~720 ± 1) Torr of synthetic air and SPME/GC-TOFMS at atmospheric pressure.
During the reaction process in the 50 L Pyrex® glass chamber, the identification of products was made using the
FTIR analysis but, at the same time, a sample was taken and analyzed in the SPME/GC-TOFMS system.  In
addition, independent experiments using SPME/GC-TOFMS technique in a 150/500 L Teflon® reactor were
developed. Products analyses were carried out using the same procedure as for the kinetic experiments, without
the reference compound, and employing synthetic air as bath gas. In this occasion the heating of the oven was
changed slightly in order to get a better separation and to detect the products generated. The temperature ramps of
the oven employed in the chromatograph were: 40 ℃ for 4 min; ramp, 25 ℃ min$^{-1}$ to 120 ℃, held for 10 min,
second ramp, 25 ℃ min$^{-1}$ to 200 ℃, held for 4 min.
The qualitative analysis in the FTIR experiments was carried out using the FTIR library that provides the FTIR
spectrophotometer          (Aldrich          vapor          phase          sample          library,
https://www.thermofisher.com/search/browse/results?customGroup=Spectral+Libraries) and/or the FTIR
database of Eurochamp (https://data.eurochamp.org/data-access/spectra/ ).
For the SPME/GC-TOFMS experiments, the NIST webbook (https://webbook.nist.gov/chemistry/) and the mass
spectra database of the instrument were used to identify the products. Calibrated FTIR spectra and SPME/GC-
TOFMS chromatograms of authentic samples were used in those case where the product is commercially available.
The yields of the reaction products were estimated from the slopes of plots of the concentration of formed product
versus the amounts of MSA (Δ[MSA]) consumed. To obtain the yield in percentage of carbon, the yield obtained
is multiplied by 100 and by the ratio of carbons between the product and the MSA from which it comes.
Sometimes, where important loss of reaction product could be taken on by reaction of the oxidant and by photolytic
process, the concentration of product was corrected using the formulism of Tuazon et al. (1986) (see S1 in
supplementary material). Range concentrations of reactants employed were as follows: 2-14 ppm of MSA, 8-31
ppm of Cl$_2$, 12-57 ppm of NO, 19-66 ppm of methylnitrite and 6-36 ppm of N$_2$O$_5$.
Chemicals used were as follows: 4MCHexOH (97 %, Aldrich), 3,3DM1ButOH and 3,3DM2ButOH (98 %,
Aldrich); 1-butene, propene, 2-methyl-2-butanol, isopropanol, 2-methylpropene, 4-methylcyclohexanone and
cyclohexene ( ≥ 99 %, Aldrich), 2-ethyl-1-hexanol ( ≥ 99 %, Fluka), 1-butanol (99.8 %, Aldrich), 3,3-
dimethylbutanal (95 %, Aldrich) and 3,3-dimethyl-2-butanone (98 %, Aldrich), NO (99 %, Praxair), Cl$_2$ ( > 99.8
%, Praxair), synthetic Air (Praxair Ultrahigh purity 99.999 %,), N$_2$ (99.999 %, Praxair). For N$_2$O$_5$ synthesis, N$_2$O$_4$
were used (>99.5 %) from Fluka, P$_2$O$_5$ (98 %, such as desiccant) from Fluka and O$_3$ synthesized by a generator
model TRCE-5000, 5 g$_{O3}$ h$^{-1}$ OZOGAS.
**3. Results and discussion**
**3.1 Kinetic study**
Preliminary test experiments indicated that dark heterogeneous reactions and photolytic losses of MSA can be
considered negligible (k ~ 10$^{-6}$ s$^{-1}$). As mentioned above, the kinetic study of Cl atoms and OH radical with the
organics was carried out at room temperature (~ 298 K) and at ~720 Torr of N$_2$ gas and synthetic air respectively.





Nitrate radical experiments were performed at room temperature and atmospheric pressure using $N_2$ gas in a 500
L Teflon bag and employing the system SPME/GC-TOFMS. A number of injections of the unreacted mixture
were carried out in order to determine the associated precision with the sampling method to be used in the error
analysis (Brauers and Finlayson-Pitts, 1997). The standard deviations (σ) were as follows: 3.7 % for 4MCHexOH,
1.7 % for 1-butanol and 3.5 % for 2-ethyl-1-hexanol. Figure 1 shows examples of the kinetic data plotted according
to Eq (1) for the reactions of MSA with different atmospheric oxidants.
A good correlation was obtained with an intercept close to zero, which indicates the absence of other secondary
processes. From the slopes of the plots ($k_{MSA}/k_R$) and known values for the rate coefficients for the reference
compounds employed ($k_R$) the value of the absolute rate coefficient for each methyl saturated alcohol ($k_{MSA}$) has
been determined.
Rate coefficients of reference compounds, for Cl atom reactions (in $10^{-10}$ cm$^3$ molecule$^{-1}$ s$^{-1}$ units) were: 2-
methylpropene (3.40 ± 0.28), 1-butene (3.38 ± 0.48), (Ezzel et al., 2002) and propene (2.23 ± 0.31) (Ceacero-Vega
et al., 2009); for OH radical reactions (in $10^{-11}$ cm$^3$ molecule$^{-1}$ s$^{-1}$ units): propene (2.66 ± 0.40), (Atkinson and
Aschman, 1989), cyclohexene (6.77 ± 1.69) (Atkinson and Arey, 2003), isopropanol (0.51 ± 0.008) (IUPAC
www.iupac-kinetic.ch.cam.ac.uk) and 2-methyl-2-butanol (0.36 ± 0.06) (Jiménez et al., 2005). And for $NO_3$
reactions (in $10^{-15}$ cm$^3$ molecule$^{-1}$ s$^{-1}$ units): 1-butanol (3.14 ± 0.97) and 2-ethyl-1-hexanol (2.93 ± 0.46) (Gallego-
Iniesta et al., 2010). The experimental data are shown in Table 1. The rate coefficients obtained in this work are
the first kinetic data reported for these MSA, therefore results obtained can not be compared with literature values.
As it has been mentioned in introduction section, it is well stablished that the gas-phase reaction mechanism of
saturated organic compounds (alkanes, alcohols, ethers, etc) with the atmospheric oxidants (Cl atoms, OH and
$NO_3$ radicals) are initiated "via" hydrogen atom abstraction from the organic compound to form a stable molecule
and an alkyl radical (Finlayson-Pitts and Pitts, 2000; Atkinson and Arey, 2003; Calvert et al., 2011; Ziemann and
Atkinson, 2012). The presence of hydroxyl group in saturated alcohols implies two types of hydrogens that can be
subtracted, hydrogen bonded to carbon (C-H) of main chain or to an alkyl substituent and hydrogen bonded to
oxygen of hydroxyl substituent (-OH). Two literature reviews about reactivity of saturated alcohols (Calvert et al.,
2011; Mellouki et al., 2015) conclude that:
1- The reactions of aliphatic alcohols with atmospheric oxidants proceed mainly by H atom abstraction from
various C−H groups in the alkyl chain being the H atom abstraction from the O−H negligible.
2-Rate coefficients for the reactions of Cl, OH and $NO_3$ are higher than those of the corresponding alkanes due to
the activating effect of the OH group. This effect is extended over about 4 carbon atoms (Nelson et al., 1990). As
will be discussed later, the activating effect of the OH group is depending on the oxidant.
3-The attack percentage of radical to the different sites of alcohol (α, β, γ and δ) depends on the oxidants, structure
of saturated alcohol, type, numbers of substituents and temperature. (Moreno et al., 2012, 2014; McGillen et al.,

35   2013, 2016).

Taking into account these remarks, the reactivity of the Methyl Saturated Alcohols studied in this work will be
analyzed and discussed comparing the rate coefficients of these MSA with: a) the different oxidants, b)  the same
oxidant with different alcohols and c) with the rate coefficients of their homologous alkanes. The data used to
compare are summarized in Table S1 in supplementary material.
From the analysis of data, it can be observed that:





1-The trend in the reactivity of MSA in relation to the different oxidants is the same that the observed for other
saturated alcohols: $k_{Cl}$ (k ~ $10^{-10}$) > $k_{OH}$ (k ~ $10^{-11}$) >> $k_{NO3}$(k ~ $10^{-15}$), (k in cm$^3$ molec$^{-1}$ s$^{-1}$ units). This behaviour
could be explained for the different size and electronic properties of each oxidant that make the Cl atom the most
reactive (value of k in the limit of collision) but also less selective than OH and NO$_3$ radicals.
2- The rate coefficient for the reaction of 4MCHexOH with Cl atoms is similar to the rate coefficient of its
homologous alkane (E-1,4-dimethylcyclohexane): $k_{4MCHexOH}$ =37.1 × $10^{-11}$ ≅ $k_{E\text{-}1,4\text{-}dimethylcyclohexane}$ =36.3 × $10^{-11}$. In
the case of the reaction with OH radical, the rate coefficient of 4MCHexOH is 1.6 times higher than E-1,4-
dimethylcyclohexane (see data of Table S1). These results show that the activating effect of hydroxyl group is less
important for the Cl reactions due to the high reactivity of Cl atoms. In the case of 3,3-dimethylbutanols, there is
not data of rate coefficients of the homologous alkanes for comparison, but in general it is observed a large
influence of the structure of the organic compound on the reactivity (SAR Method, Kwok and Atkinson, 1995).
This effect has been quantified for each of the functional groups of an organic compound. So, in the case of alcohols
the factor of hydroxyl group, is 1.18 for the reaction with Cl, 2.35 for the reactions with OH (Calvert et al., 2011)
and 18 for the reactions of the nitrate radical (Kerducci et al., 2014).
3-Rate coefficients obtained for these three MSA with the same oxidant are of the same order that the
corresponding to other saturated alcohols (See data of Table S1). The activating effect of the length chain in the
reactivity is being more marked in the Cl reaction than in the case of OH and NO$_3$ reactions. Again, this behavior
could be explained by the different order of reactivity between the oxidants. For Cl atom, more reactive but less
selective, an increase of chain implies more hydrogens available to be subtracted and therefore an increase of the
rate coefficient. However, the OH and NO$_3$ radicals, less reactive and more selective, the attack to subtract the
hydrogen will be carried out in a specific place, so an increase of the chain doesn't affect the reactivity
significantly.
In the reactions of OH and NO$_3$ radicals, the presence of activating substituents or the formation of a more stable
radical after the H-abstraction could have a major effect in the reactivity than in the case of Cl atom reaction. This
last assumption could also explain the minor rate coefficient observed for the Cl reaction with secondary alcohols
(2-propanol, 2-butanol, 2-pentanol, 3-methyl-2-butanol and 3,3-dimethyl-2-butanol) instead of primary alcohols
(1-propanol, 1-butanol, 1-pentanol, 3-methyl-1-butanol and 3,3-dimethyl-1-butanol) (see Table S1).
All that could imply a different mechanism in the hydrogen abstraction process for Cl atoms versus OH and NO$_3$
radicals. Nelson et al. (1990) and Smith and Ravishankara (2002) indicate the possible formation of an
intermediate adduct between the OH radical and the oxygen of the hydroxyl group via hydrogen bond that will
imply a specific orientation. Theoretical studies found in bibliography show this different hydrogen-abstraction
process in the reaction of saturated alcohols with Cl atoms (Garzon et al., 2006) and OH (Moc and Simmie, 2010).
These differences in the mechanism for each oxidant should be observed in the analysis of the reaction products
implying different yields and products distributions.
**3.1.1 Estimation of rate coefficients**
In order to estimate the rate coefficient of the reactions of organic compounds with the atmospheric oxidants,
multitude of methods have been proposed (Vereecken et al. 2018). The most popular and used is the SAR method





develop initially by Kwok and Atkinson (1995) to estimate the rate coefficients at room temperature for gas phase
reactions of OH radical. This method has been updated for OH reactions (Jenkin et al., 2018) and extended to $NO_3$
(Kerducci et al., 2010, 2014) and Cl (Calvert et al., 2011; Poutsma 2013) reactions. The EPA (United States
Environmental Protection Agency) has developed the EPI Suite™-Estimation Program Interface that allows to
estimate the rate coefficient for the reaction of OH radical and organic compounds using the AOPWIN v1.92
program. In this work the rate coefficients of MSA with the three oxidants have been estimated using the SAR
method and are shown in Table 2.
The values of estimated rate coefficients agree with experimental data with ratios $k_{exp}/k_{SAR}$ between 0.8 and 1.27,
except for the case of 3,3DM1ButOH and $NO_3$ radical with a $k_{exp}/k_{SAR}$ of 3.29. In general, the SAR method applied
to alcohols predicts better rate coefficients for Cl atoms and OH radical than for $NO_3$ radical, especially for primary
alcohols. It is important to note that the kinetic database for the $NO_3$ reactions is more limited than for Cl and OH
reactions, so the estimated rate coefficient for $NO_3$ radical should be treated with caution (Calver et al., 2011).
It is known that organic compounds which reacts in the same way with different atmospheric oxidants, present a
correlation between their rate coefficients. In this sense, along the years, different correlations have been proposed
that allow to estimate the unknown rate coefficient when the other one is known (Wayne, 1991, 2000; Atkinson,
1994; Calvert et al., 2011; Gallego-Iniesta et al., 2014). Correlations $logk_{Cl}$-$logk_{OH}$ and $logk_{NO3}$-$logk_{OH}$ have been
built for a set of alcohols, ethers and saturated alcohols by Calvert et al., (2011) obtaining the following
relationships:
$$log(k_{Cl}/cm^{-3}molecule^{-1}s^{-1})=0.634\times log(k_{OH}/cm^{-3}molecule^{-1}s^{-1})-2.71 \qquad (r^2=0.72) \qquad (2)$$
$$log(k_{NO3}/cm^{-3}molecule^{-1}s^{-1})=1.11\times log(k_{OH}/cm^{-3}molecule^{-1}s^{-1})-2.42 \qquad (r^2=0.66) \qquad (3)$$
These equations have been used to estimate the rate coefficients of the reactions of MSA with Cl and nitrate radical
using the experimental rate coefficients measured in this work for OH reactions. The results obtained are (k in $cm^3$
$molecule^{-1}$ $s^{-1}$ units): $k_{Cl-3,3DM1ButOH}=14.3 \times 10^{-11}$; $k_{Cl-3,3DM2ButOH}=21.4 \times 10^{-11}$; $k_{Cl-4MCHexOH}=31.2 \times 10^{-11}$; $k_{NO3-}$
$_{3,3DM1ButOH}=1.22 \times 10^{-15}$; $k_{NO3-3,3DM2ButOH}=2.48 \times 10^{-15}$ and $k_{NO3-4MCHexOH}=4.81 \times 10^{-15}$. This estimation method
obtains slightly better results than SAR for $NO_3$ reactions. The better prediction for the $NO_3$ rate coefficients than
for those of Cl could be due to the fact that the mechanism for Cl atom reactions is different than for OH radical
reaction. Assumption that must be satisfied to apply the correlation. It is important to indicate that in the case of
Cl reactions, other effects as thermochemistry and the polar effect, must be considered to estimate rate coefficients
for hydrogen abstraction reactions (Poutsma, 2013).
**3.2 Product and Mechanistic Study**
A product study of 4MCHexOH, 3,3DM1ButOH and 3,3DM2ButOH with chlorine atoms in absence/presence of
$NO_x$, hydroxyl and nitrate radicals has been performed by employing the two experimental set-ups mentioned
above (FTIR and SPME/CG-TOFMS). IR absorption bands of HCl, $CO_2$, CO, $HNO_3$, $N_2O$, $NO_2$, HCOOH, HCOH,
ClNO, $ClNO_2$ and $CH_3NO_3$ were observed in the FTIR experiments. Bands that are due to the decomposition of
the precursors employed ($Cl_2$, $CH_3NO_2$ and $N_2O_5$) and in some cases due to heterogeneous reactions with the walls
of the gas cell. The formation of $O_3$ and $N_2O_5$ have also been observed at large reaction time for reactions of MSA
and Cl atoms in presence of NO, due to the high concentration of the $NO_2$ in the medium of reaction. Quantitative



analysis was carried out by linear subtraction of a spectrum's absorption bands and peak areas of GC
chromatograms with the use of calibrated spectra and reference chromatograms.
The experimental conditions and yields of the main products formed in the reactions of MSA and analyzed by
FTIR and SPME/GC-TOFMS techniques are given in Tables 3-6. Yields could be affected to large errors
associated with the SPME sampling method and due to the presence of interfering IR band absorptions, mainly
associated with precursors of OH and NO$_3$ radicals or by nitrated compounds formed.
**3.2.1 - 4MCHexOH**
E-4-methylcyclohexanone was identified in the reaction with Cl, Cl+NO, OH + NO and NO$_3$ + NO$_2$. An example
of the product spectra obtained in the FTIR system is shown in Fig. 2. Formation of the E-4-methylcyclohexanone
was confirmed by introducing a sample of the commercial product (spectrum (e)). A set of experiments using the
SPME/GC-TOFMS system were also carried out for the reaction of 4MCHexOH with Cl atoms and OH and NO$_3$
radicals. An example of the chromatogram obtained for the reaction of 4MCHexOH with chlorine atoms is shown
of Fig. 3. In all the studied reactions, formation of a product peak at 10.35 min was observed. This peak (B) on
Fig. 3 was assigned to E-4-methylcyclohexanone and confirmed by comparing with the retention time and MS
spectrum of a commercial sample. In the reactions with chlorine atoms (absence/presence of NO$_x$) and OH radical
two additional peaks at 19.80 min (C) and 20.25 min (D) were observed.
The time-concentration profiles of 4MCHexOH and E-4-methylcyclohexanone obtained by FTIR for the reaction
with chlorine atoms in the presence of NOx is shown on Fig. S1. The concentrations of E-4-methylcyclohexanone,
corrected according to Eq (S1), (S2) and (S3), were plotted versus the amounts of 4MCHexOH consumed in order
to obtain the yield of 4-methylcyclohexanone from the slope.  An example of the obtained plots is shown in Fig.
4. Yields, Y (%), of E-4-methylcyclohexanone obtained in all experiments are listed in Table 3. Based on the
average yield of E-4-methylcyclohexanone, the carbon balance is below to 50 % for reactions with Cl and OH
radical and ~ 60 % for NO$_3$.
Residuals spectra after subtraction of E-4-methylcyclohexanone show IR absorption bands compatible with the
presence of hydroxy carbonyl compounds (~1750, 1720, 1060 cm$^{-1}$) and nitrated organic compounds (RONO$_2$ ~
1260, 1264 and 862 cm$^{-1}$, and/or ROONO$_2$ ~ 1720, 1300 and 760 cm$^{-1}$)  (See residual spectra, Fig. S2 in
supplementary material). The amount of nitrated compounds was estimated using the average integrated absorption
coefficient of 1.2x10$^{-17}$ cm molecule$^{-1}$ of similar compounds corresponding to the IR band 1260-1305 cm$^{-1}$ (Tuazon
and Atkinson,1990). The calculated yields of RONO$_2$ were 20 % and 60 % for Cl + NO and NO$_3$ reactions
respectively. A yield of 10 % of nitrated compounds was estimated for the reaction with OH radical. This lower
yield could be due to fact that the NO$_x$, present in the reaction medium, reacts faster with the CH$_3$O$^\bullet$ (formed in
the reaction of CH$_3$ONO with NO) than others alcoxyradicals. Table 6 shows a summary of the average yields of
reaction products quantified for 4MCHexOH.
Considering the products detected here and the detected in the study of Bradley et al., (2001) relative to
cyclohexanol with OH radical reactions, a degradation mechanism for 4MCHexOH with the atmospheric oxidants
has been proposed. Figure 5A shows the paths to explain the formation of carbonyl or hydroxy carbonyl



compounds and Figure 5B shows an example path to explain the formation of nitrated organic compounds
($ROONO_2$ and $RONO_2$). Similar compounds could be formed by routes II-IV. The abstraction of Hydrogen atoms
in α-position with respect to alcohols group (channel I) followed by the addition of oxygen, formation of a peroxy
radical and fast decomposition of this radical explains the formation of E-4-methylcyclohexanone. Based on the
yield obtained for E-4-methylcyclohexanone for each oxidant (See Table 6), this channel represents ~ 25-30 %,
~40 % and ~60 % of reaction mechanism of 4McHexOH with Cl and Cl+NO, OH and $NO_3$ reaction, respectively.
Percentages are two times higher than SAR method prediction in the case of Cl atoms reactions and 1.3 times
higher for OH and $NO_3$ reactions. It should be noted that these data should be taken with caution, since they could
imply many sources of error.
Apart from E-4-methylcyclohexanone, other carbonyl and hydroxy carbonyl compounds could be formed by
routes II, III, and IV. The presence of this kind of compounds have been observed in the reactions with Cl and OH.
According to the EI MS spectra (Fig. S3, supplementary material) of peaks (C) and (D) shown in Fig. 3, an
assignation to 2-hydroxy-5-methyl-cyclohexanone and 5-hydroxy-2-methyl-cyclohexanone or 3-methyl-1,6-
hexanedial respectively has been proposed. However, according to the atmospheric reactivity (Finlayson and Pitts,
2000; Calvert et al., 2011; Ziemann and Atkinson, 2012) and the study of Bradley et al., (2001), the compound
that would be expected is 3-methyl-1,6-hexanedial, which comes from the decomposition of the alkoxy radical
formed in route II. However, confirmation was not possible since these compounds are not commercially available.
The detection of HCOH about 9 % in the reaction with Cl atoms indicates that the elimination of the methyl group
in route IV is minor.
In the case of nitrate radical only E-4-methylcyclohexanone was detected as carbonyl compound, suggesting that
the route I may be the dominant pathway for this radical. The large difference between the yields of E-4-
methylcyclohexanone obtained using the SPME/GC-TOFMS system (~75 %) or the FTIR (35 %) could be due to
the influence of the volume of reactor (150 or 500 liters in the SPME/GC-TOFMS compared to 50 L of the FTIR),
which favors the formation of carbonyl compounds instead of nitrates in the case of using a large volume reactor.
This fact is more pronounced in the case of reactions with nitrate radical since, due to the precursor used, the
reaction occurs in the presence of high concentrations of $NO_2$ favoring the addition of $NO_2$ to peroxy or alkoxy
radicals (See Figure 5B). Taking into account, the yields of E-4-methylcyclohexanone and the nitrated compounds
for the $NO_3$ reaction using FTIR, a total carbon balance of 100% is obtained (See Table 6).
**3.2.2 -3,3DM1ButOH**
Following the same procedure as above, 3,3-dimethylbutanal was identified as the main reaction product in the
reaction of 3,3DM1ButOH with the three atmospheric oxidants. Figure S4A shows the FTIR spectra obtained for
the reactions of 3,3DM1ButOH with Cl, Cl + NO, OH and $NO_3$ after subtraction.
Residual FTIR spectra after subtraction of 3,3-dimethylbutanal (Fig. S4B), the SPME/GC-TOFMS
chromatograms (Fig. S5) and EI MS spectra (Fig. S6), show that other reaction products as carbonyl, hydroxy
carbonyl and nitrated compounds are formed. These compounds could be HCOH, 2,2-dimethylpropanal,
glycolaldehyde, acetone, peroxy-3,3-dimethyl-butyryl nitrate (P33DMBN) $(CH_3)_3CCH_2C(O)OONO_2$. These
compounds can be formed as primary products (see Fig. 6 below) or secondary products from degradation of 3,3-



dimethylbutanal (See Fig. S7). The SPME/GC-TOFMS chromatograms show common peaks for the three
oxidants, but the number of peaks and its distribution are very different, especially for OH reactions. In the case
of SPME/GC-TOFMS system a set of experiment using Field Ionization was carried out in order to help us to
stablish the identification of reaction products.
Time-concentration profiles of 3,3DM1ButOH, 3,3-dimethylbutanal and those reaction products positively
identified by FTIR analysis were made in order to stablish if the profiles correspond with a primary or secondary
reaction products. An example of the reactions with chlorine atoms in the absence and presence of NOx is shown
on Fig. 6, observing that in the absence of NO the profiles of acetone and formaldehyde show a typical profile of
secondary reactions. This behaviour is only clearly observed in the profile of nitrated compounds in the reaction
of Cl atoms in the presence of NO (Fig. 6B).
Commercial sample of 3,3-dimethylbutanal was used to estimate yields in both experimental systems. These yields
are shown in the Table 4. The yields of acetone and HCOH were calculated using a FTIR reference spectrum of
commercial sample and FTIR reference spectra from Eurochamp database (https://data.eurochamp.org/data-
access/spectra/), respectively. A FTIR reference spectrum of 2-methylpropanal (from this same database) has been
used to estimate the yield of 2,2-dimethylpropanal.
The amounts of 3,3-dimethylbutanal formed were corrected by its reaction with Cl atoms, and OH and $NO_3$ radicals
as is described previously using the rate coefficients available in bibliography or for reactions of structurally
compounds similar (see footnote Table 4). Estimated yields of formaldehyde, acetone, 2,2-dimethylpropanal and
nitrated compounds are summarized in Table 6 together with an average yield of 3,3-dimethylbutanal. The higher
yield of nitrated compounds in the reaction of 3,3DM1ButOH with nitrate radical could indicate secondary
products (See Fig. S7). A total of carbon yield (nitrated compounds have not been accounted) of ~70 %, ~85 %
and ~40 % have been justified for Cl (absence and presence of NO), OH and $NO_3$ reaction respectively, but must
be noted that there are reaction products that could not possibly be quantified as carbonyl or hydroxy carbonyl
compounds in the Cl atoms reactions and primary nitrated compounds in the Cl + NO and $NO_3$ radical reactions.
This work is the first study of reaction products of 3,3DM1ButOH with the atmospheric oxidants, so there is not
any study to compare. Figure 7 shows the reaction mechanism proposed based on the literature studies about
saturated alcohols reactions with Cl atoms and OH radical (Cavalli et al., 2002; Hurley et al., 2009; McGillen et
al., 2013; Welz et al., 2013) and considering the reaction products identified in this work. Table S2, in
supplementary material, shows a summary of the reaction products proposed in this mechanism observed or
tentatively identified in the reactions of 3,3DM1ButOH with the atmospheric oxidants.
Estimated yields of 3,3-dimethylbutanal (formed by H-Abstraction in α position of 3,3DM1BuOH) for Cl and OH
reaction (~(40-43) %, ~57 %) are very similar to the one predicted by the SAR method (40 % and 66 %
respectively). In the case of $NO_3$ radical a large difference between both yields are observed (33 % estimated in
this work, 85 % predicted by SAR method). This discrepancy could be explained by the fact that the SAR method
(Kerducci et al., 2014) underestimates the attack of $NO_3$ in β-position, because it does not consider the possible
effect of the alcohol group jointed to $-CH_2$. That is, the SAR method considers the effect of $-CH_2$ and perhaps must



also consider the factor of -CH$_2$OH. This could also explain the large difference observed between the estimated
and measured rate coefficient as was shown in Table 2. On the other hand, as discussed above, the volume of the
reactor can also have influence to formation of 3,3-dimethylbutanal.
According with the yields of the products quantified and/or observed in the SPME/GC-TOFMS chromatograms,
it can be concluded that for OH radical reaction, the route I (attack in α position) seems to be the main reaction
route. For Cl atoms the three routes can occur although the I and III (attack in δ position, especially evident in
presence of NO) seem to be the major routes. The major yield of HCOH, acetone and 2,2-dimethylpropanal in the
reactions of Cl atoms in presence of NO versus Cl atoms reactions in absence of NO could indicate that in absence
of NO the self peroxy radical reaction "via" molecular channel (formation of carbonyl and dihydroxy organic
compounds) is more favored than "via" radical channel, with formation of two alcoxy radicals. For NO$_3$ radical,
routes I and II (attack in β position) with formation of nitrated compounds apart from 3,3-dimethylbutanal seem
to be the unique routes.
**3.2.3 -3,3DM2ButOH**
The analysis of FTIR spectra obtained for the reactions of 3,3DM2ButOH with Cl atoms, in presence and absence
of NO, OH radical and NO$_3$ radical shows the formation of 3,3-dimethyl-2-butanone as a main product (see Fig.
S8). Others compounds such as HCOH, acetone, 2,2-dimethylpropanal and Peroxy Acetyl Nitrate (PAN), have
also been observed. The residual FTIR spectra after substraction of all known IR bands again shows the presence
of carbonyl compounds (IR bands absorption in the range of 1820-1700 cm$^{-1}$); hydroxy compounds (1060-1040
cm$^{-1}$) in the reaction of Cl atoms in absence of NO and also nitrated compounds (RONO$_2$; 1650, 1305-1260, 890
cm$^{-1}$) in the reaction of Cl + NO and NO$_3$ radical (Fig. S8C). The presence in the residual FTIR spectra of a IR
absorption band around 1800 cm$^{-1}$ in the reaction of Cl atoms at large reaction times could be due to the formation
of chlorine compounds by reaction of 3,3dimethyl-2-butanone with Cl$_2$ (Ren et al., 2018) or the formation of cyclic
compounds as hydrofurans. The SPME/GC-TOFMS chromatograms and MS spectra (Fig. S9 and S10) confirm
other reaction products apart from 3,3-dimethyl-2-butanone in the case of Cl, Cl + NO and NO$_3$ reactions. Only
one peak is observed in chromatograms obtained for the OH reactions.
Acetone, HCOH, 2,2-dimethylpropanal, nitrated compounds and acetaldehyde have also been quantified. Plots of
concentration versus time show, typical profiles of secondary reactions for HCOH, acetone and nitrated
compounds in the reactions of Cl with NO (See Fig. S11). These compounds could also be formed by degradation
of 3,3dimethyl-2-butanone (See Fig. S12). The estimated yields of 3,3-dimethyl-2-butanone for all individual
experiments are given in Table 5, where the measured concentrations have again been corrected for secondary
reactions. Table 6 summarizes the yields of all quantified products.
A total carbon yields of ~60 %, 112 %, 95 % and 58 % have been accounted for Cl (absence and presence of NO),
OH and NO$_3$ reaction, respectively (See Table 6). It is important to note that in the case of the reaction of Cl atoms
without NO where the total carbon yield is lower than 100 % there are many reaction products that could not be
quantified, as carbonyl and/or hydroxy carbonyl compounds. In the reaction of NO$_3$ radical, due to our
experimental conditions, an important amount of primary nitrated compounds is expected to be formed (See Fig.
S8C).
A mechanism of hydrogen abstraction in different positions of the carbon chain has been proposed for the reaction
of 3,3DM2ButOH with Cl, OH and $NO_3$ reactions. The mechanism is shown in Figure 8. Table S3, in
supplementary material, shows a summary of the reaction products proposed in this mechanism observed or
tentatively identified in the reactions of 3,3DM2ButOH with the atmospheric oxidants.
Yields of 3,3-dimethyl-2-butanone obtained in this work imply a percentage of attack of the oxidant in β position
(route I of mechanism) of: 43 % and 44 %, in the case of chlorine atom; 81 % for OH radical and 58 % for $NO_3$
radical. Percentages are very similar to that predicted by SAR method except for $NO_3$ radical. High $NO_2$
concentration present in the reaction would highly favor the formation of nitrated compounds versus 3,3-dimethyl-
2-butanone. The main reaction products observed in the reaction of Cl atoms in presence of NO (3,3-dimethyl-2-
butanone, HCOH, 2,2-dimethylpropanal, acetone, acetaldehyde) confirm that the Cl atoms could attack in other
sites with an important percentage. Based on the estimated yield of acetone, the attack in δ position with abstraction
of hydrogen of methyl groups (route III) could be ~58 %, and based on the estimated yield for 2,2-
dimethylpropanal, the route (I) could account with a 10 %. These data agree with the SAR predictions for Cl atom
reactions. On the other hand, the major yields of acetone, HCOH, 2,2-dimethylpropanal and acetaldehyde in the
reaction of Cl atoms in presence of NO than in absence of NO could indicate that in presence of NO the peroxy
self-reactions ($RO_2$) "via" molecular channel is negligible. The lower yield (17 %) estimated by acetaldehyde
versus 58% of its coproduct (acetone) is due to its fast degradation by Cl atoms reaction with formation of Peroxy
acetyl nitrates as it has been observed in the FTIR experiments (See Fig. S8B).
**4. Atmospheric Implications**
The pollutants in the atmosphere, could create serious environmental problems such a photochemical smog, acid
rain and degradation of the ozone layer (Finlayson-Pitts and Pitts, 2000). So, it is important to evaluate the
parameters that help us to know the impact of the presence of these compounds in the atmosphere. These
parameters are, the time that such compounds remains in the atmosphere, the Global Warming Potencial (GWP)
and their degradation mechanisms in order to estimate the impact of products formed.
The first important parameter of the environmental impact of an Oxygenated Volatile Organic Compounds in the
atmosphere, is the global lifetime, $\tau_{global}$, which considers all the degradation processes which could suffer these
compounds in the Troposphere. This parameter can be obtained from the sum of the individual sink processes such
as reactions initiated by OH and $NO_3$ radicals, Cl atoms, and $O_3$ molecules; photolysis and dry and wet deposition,
Eq (4):
$$\tau_{global} = \left[ \frac{1}{\tau_{OH}} + \frac{1}{\tau_{Cl}} + \frac{1}{\tau_{NO_3}} + \frac{1}{\tau_{O_3}} + \frac{1}{\tau_{photolysis}} + \frac{1}{\tau_{other\ processes}} \right]^{-1} \quad (4)$$
Tropospheric lifetime (τ) of 4MCHexOH, 3,3DM1ButOH and 3,3DM2ButOH for each process have been
estimated considering Eq (4) and (5).
$$\tau = \frac{1}{k_{Ox}[Ox]} \qquad (5)$$





where $k_{OX}$ and [Ox] are the rate coefficient obtained in this work for each oxidant and typical atmospheric
concentration of the oxidants Cl, OH and NO$_3$, respectively. Concentrations employed were as follows: for 24
hours average: $1 \times 10^3$ atoms cm$^{-3}$ (Platt and Janssen, 1995) for chlorine atoms, 12-hours average day-time
concentration of $1 \times 10^6$ radicals cm$^{-3}$ for OH (Prinn et al., 2001) and $5 \times 10^8$ radicals cm$^{-3}$ for NO$_3$ radicals
(Atkinson, 2000), and a peak concentration of chlorine atoms of $1.3 \times 10^5$ atoms cm$^{-3}$ in the coastal marine
boundary layer at dawn (Spicer et al., 1998). Reactions with O$_3$ and photolysis are negligible loss processes for
this kind of compounds (Mellouki et al., 2015). Other processes are referred to dry a wet deposition. For estimating
the lifetime associated with wet deposition, Eq (6) proposed by (Chen et al. 2003) has been used:
$$\tau_{wet} = \frac{H_{atm}}{v_{pm}RTk_H} \quad (6)$$
Where $k_H$ is the Henry's law constant, $H_{atm}$ is the height in the troposphere taking a value of 630 m, $v_{pm}$ is the
average precipitation rate for Ciudad Real (Spain) (402 mm/year) (www.aemet.es), R is the gases constant and T
is the temperature considered as constant and equal to 298 K. In bibliography there is only data of the constant of
Henry for 3,3DM2ButOH ($5.6 \times 10^{-1}$ mol m$^{-3}$ Pa$^{-1}$) (Sander, 2015). Comparing the available data for similar
compounds it has been used an approximated value of $K_H$ of 3 mol m$^{-3}$ Pa$^{-1}$ and 0.4 mol m$^{-3}$ Pa$^{-1}$ for 4MCHexOH
and 3,3DM1ButOH respectively.
Lifetime calculated of the three studied alcohols in this work are shown in the Table 7. It can be seen that the
dominant tropospheric loss process for the three alcohols is clearly their reaction with OH radicals followed by
their reaction with NO$_3$ radicals at night. However, in places where there is a peak concentration of chlorine atoms
(coastal areas) the reaction of these alcohols with chlorine atoms may compete with OH radicals becoming their
main degradation process.
The global lifetime of the three alcohols is of the order of ~ 1-2 days, indicating that these compounds will probably
be degraded near their sources. These global lifetimes also indicate that MSA have not a significant contribution
to radiative forcing of climate change (Mellouki et al., 2015), which is supported through the estimation of their
GWP values. For time-horizon of 20 years, the values estimated have been: $8.33 \times 10^{-4}$, $1.78 \times 10^{-2}$ and $5.80 \times 10^{-3}$
for 4MCHexOH, 3,3DM1ButOH and 3,3DM2ButOH respectively, which are very low. So, these compounds
will only have an important impact in the troposphere at local or regional level.
Their degradation products (mostly carbonyl-containing compounds and nitro-compounds in polluted areas) must
be considered. Thus, the nitrated compounds generated can act as NOx reservoir species especially during the
night (Altshuller, 1993) and could have influence at global scale. Moreover, since 4MCHexOH, 3,3DM1ButOH
and 3,3DM2ButOH react quickly with chlorine atoms and OH radicals, their contribution to the formation of
photochemical smog might be important. For that reason, the contribution of these three alcohols to the formation
of smog was estimated by obtaining the average ozone production during 99 % of their reactions with OH radical,
using the equation indicated by Dash and Rajakumar (2013). The values obtained were 3.24, 0.90 and 1.69 ppm
for 4MCHexOH, 3,3DM1ButOH and 3,3DM2ButOH, respectively. These values suggest that these compounds
may be a potential generators of tropospheric ozone and could contribute significantly to the formation of
photochemical smog (depending on their environmental concentration).
**5. Conclusions**
The main conclusions that have been obtained with the present study, are the following:





-The kinetic and product study confirms that the atmospheric degradation mechanism for methyl saturated
alcohols and possibly for the rest of unstudied saturated alcohols, proceeds mainly by abstraction of the hydrogen
atom bonded to carbon instead hydrogen atoms bonded to oxygen atom of the alcohol group.
-Chlorine atoms subtract any type of hydrogen from saturated alcohols with a high percentage, compared to the
hydroxyl radical and the nitrate radical. OH and $NO_3$ radicals subtract mainly the hydrogen in the $\alpha$ position, if
the saturated alcohols are secondary. For primary alcohols the hydrogen in $\alpha$ position is subtracted almost
exclusively for the OH radical and $\alpha$ and $\beta$ position to 50 % in the case of the $NO_3$ radical, extending the inductive
effect to $\alpha$ and $\beta$ position. Therefore, for the reaction of $NO_3$ radical, it is necessary to update the SAR method
developed by Kerducci et al., (2010, 2014) to take into account the effect of the OH group in $\beta$ position, (-
$CH_2OH$) and not only the effect in $\alpha$ position (-OH).
-Theoretical ab-initio studies should be done in order to obtain more information about how the type of radical
determine the distribution of reaction products specially in reactions that occurs by an initial hydrogen
abstraction, that is the case of primary methyl saturated alcohols.
-The atmospheric conditions determine the reaction mechanism and therefore the reaction products obtained in
the degradation of methyl saturated alcohols. So, in polluted environments with high concentrations of NOx, the
peroxyradicals ($RO_2\cdot$) reacts mainly with NO to form the alkoxy radical instead of other peroxyradical. In these
conditions, nitrated organic compounds ($RONO_2$) are formed apart from polyfunctional organic compounds.
Also, when the concentration of $NO_2$ is higher than NO concentration, ozone is formed. In clean atmosphere, as
in the case of the experiments of Cl atoms in absence of NO, the reaction products are different because of
peroxyradicals ($RO_2\cdot$) react mainly "via" self-reaction molecular channel instead to "via" self-reaction radical
channel.
-The uncounted polyfunctional could explain the low carbon balance obtained in Cl or $NO_3$ reaction. However,
the carbon balance must be taken with caution since the calculated yields have a high degree of uncertainty.
-Calculated lifetimes for methylsatured alcohols (the order of ~1 day) imply that these compounds are pollutants
at local-regional scale, but it is also important to indicate that MSA are sources of stable nitrated compounds
($ROONO_2$), depending on environment conditions, that can travel to large distance from their sources
contributing to form ozone in clean areas, for example in forest or rural areas.
-The main products coming from the degradation of the methyl saturated alcohols, aldehydes and ketones,
develop a very important secondary chemistry with formation of products of special relevance such as the PAN
observed in the degradation of 3,3-dimethyl-2-butanol. Also, more experiments should be done using other
detection techniques, in order to evaluate the formation of SOA because it is well known that polyfunctional
organic compounds are important SOA precursors.
-From the environmental point of view, this work shows that the degradation of methyl saturated alcohols is an
important source of pollutants in the atmosphere with greater or lesser impact depending on the environmental
conditions and the quantities of saturated alcohols present in the atmosphere. Therefore, the use of saturated



alcohols as additives in the production of biofuels should be controlled, avoiding that a bad handling involves
high concentrations of these alcohols in the atmosphere.
-Rate coefficients and reaction products measured in this work are the first available data, so this work contributes
to a better understanding of atmospheric chemistry of oxygenated compounds expanding the kinetic and
mechanistic data base and additionally, contributes to develop or to improve prediction models that help us to
avoid or mitigate the effects of climate change or air quality.

**6. Supplementary material.**

Attached in a separated file.

**7. Author contribution**

Salgado S. and Martín P. designed the experiments. Cabañas B. is the leader group and responsible to control the
research and got the financial support for the project leading to this publication. Colmenar I. carried out the
experiments of 4MCHexOH. Tapia A. carried out the kinetic experiments of 3,3DM1ButOH and 3,3DM2ButOH
and Aranda I. carried out the product experiments of 3,3DM1ButOH and 3,3DM2ButOH. Martín P. supervised all
analysis of data and prepared the manuscript with contributions from all co-authors.

**8. Competing interests**

The authors declare that they have no conflict of interest**.**

**9. Acknowledgment**

The authors would like to thank the financial support provided by Junta de Comunidades de Castilla-La Mancha
(Projects SBPLY/17/180501/000522).

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

| Reaction | Reference | $(k_{Cl}\pm2\sigma)/10^{-10}$ | $k_{Cl}\pm2\sigma$ /10$^{-10}$ | $\bar{k}_{Cl}\pm2\sigma$ /10$^{-10}$ |
|---|---|---|---|---|
| 3,3DM1ButOH + Cl | 1-butene | 2.89 ± 0.45 2.68 ± 0.43 2.58 ± 0.43 | 2.72 ± 0.44 | 2.70 ± 0.55 |
| | Propene | 2.63 ± 0.34 2.70 ± 0.35 2.71 ± 0.35 | 2.68 ± 0.35 | |
| 3,3DM2ButOH + Cl | 1-butene | 1.42 ± 0.31 1.17 ± 0.29 1.38 ± 0.31 | 1.32 ± 0.30 | 1.21 ± 0.37 |
| | Propene | 1.08 ± 0.22 1.12 ± 0.22 1.26 ± 0.24 | 1.15 ± 0.23 | |
| 4MCHexOH + Cl | 2-methylpropene | 3.69 ± 0.31 3.95 ± 0.31 3.35 ± 0.32 | 3.66 ± 0.31 | 3.71 ± 0.53 |
| | 1-butene | 3.86 ± 0.52 3.78 ± 0.52 3.90 ± 0.53 | 3.84 ± 0.52 | |

| Reaction | Reference | $(k_{OH}\pm2\sigma)/10^{-12}$ | $k_{OH}\pm2\sigma$ /10$^{-12}$ | $\bar{k}_{OH}\pm2\sigma$ /10$^{-12}$ |
|---|---|---|---|---|
| 3,3DM1BuOH + OH | Isopropanol | 5.09 ± 021 5.78 ± 0.47 5.72 ± 0.40 | 5.53 ± 0.36 | 5.57 ± 0.66 |
| | 2-methyl-2-butanol | 5.85 ± 0.82 5.71 ± 0.81 5.86 ± 0.83 | 5.80 ± 0.82 | |
| 3,3DM2BuOH + OH | Isopropanol | 11.90 ± 0.46 10.46 ± 0.44 9.95 ± 0.42 | 10.77 ± 0.44 | 10.51 ± 0.81 |
| | 2-methyl-2-butanol | 8.70 ± 0.98 10.64 ± 1.08 8.21 ± 0.95 | 9.18 ± 1.00 | |
| 4MCHexOH + OH | Propene | 17.1 ± 3.2 20.2 ± 3.6 20.4 ± 3.5 | 19.2 ± 3.5 | 19.1±6.5 |
| | Cyclohexene | 18.2 ± 8.8 18.4 ± 8.8 18.0 ± 8.8 | 18.2 ± 8.8 | |

| Reaction | Reference | $(k_{NO3}\pm2\sigma)/10^{-15}$ | $k_{NO3}\pm2\sigma$ /10$^{-15}$ | $\bar{k}_{NO3}\pm2\sigma$ /10$^{-15}$ |
|---|---|---|---|---|
| 4MCHexOH + NO$_3$ | 1-butanol | 3.39 ± 1.11 5.70 ± 1.82 2.51 ± 0.81 | 3.86 ± 1.25 | 2.92 ± 1.38 |
| | 2-ethyl-1-hexanol | 2.08 ± 0.72 2.93 ± 0.96 2.51 ± 0.82 | 2.51 ± 0.83 | |



The uncertainties for rate coefficients of MSA were calculated from the uncertainty of slope of plots and the
uncertainty of the reference by using the propagation of uncertainties. The average value of the rate coefficient
obtained with different reference compounds and its associated error were obtained by weighted average. $2\sigma$
statistical errors were obtained from the regression analysis ($\sigma_{slope}$) and the quoted error in the value of the rate
coefficient for the reference compound ($\sigma_{kR}$).





**Table 2.** Estimated and experimental rate coefficients ($k_{SAR}$ and $k_{exp}$) for the reaction of MSA with atmospheric
oxidants and ratio of rate coefficients ($k_{exp}/k_{SAR}$). $k$ in cm$^3$ molecule$^{-1}$ s$^{-1}$ unit.

| | Cl atoms | | | OH radical | | | NO$_3$ radical | | |
|---|---|---|---|---|---|---|---|---|---|
| | $k/10^{-11}$ | | ratio | $k/10^{-12}$ | | ratio | $k/10^{-15}$ | | Ratio |
| | $k_{SAR}$ | $k_{exp}$ | $k_{exp}/k_{SAR}$ | $k_{SAR}$ | $k_{exp}$ | $k_{exp}/k_{SAR}$ | $k_{SAR}$ | $k_{exp}$ | $k_{exp}/k_{SAR}$ |
| 4MCHexOH | 34.2[a] | 37.1[b] | 1.08 | 19.2[c] | 19.1[b] | 0.99 | 2.28[d] | 2.91[b] | 1.27 |
| 3,3DM1ButOH | 20.1[a] | 27.0[b] | 1.34 | 6.75[c] | 5.57[b] | 0.82 | 0.54[d] | 1.78[e] | 3.29 |
| 3,3DM2ButOH | 15.2[a] | 12.1[b] | 0.79 | 9.03[c] | 10.5[b] | 1.16 | 3.86[d] | 3.4[e] | 0.88 |

[a]Estimated using method described by Calvert et al., 2011
[b]Data obtained in this work
[c]Estimated using AOPWIN v1.92
[d]Estimated using method described by Kerducci et al., 2010, 2014
[e]Data obtained by Moreno et al., 2014





**Table 3.** Experimental conditions and yields of E-4-methylcyclohexanone for the reaction of
4MCHexOH with atmospheric oxidants.

| MSA | Oxidants | Exp | [MSA] (ppm) | [Precursor] (ppm) | [NO] (ppm) | Carbonyl compound [d]Yield (%) | Technique | Average[f] (%) | SAR Yield (%) |
|---|---|---|---|---|---|---|---|---|---|
| 4MCHexOH | Cl[a] | 1 | 3 | 21 | - | 24.8±0.9 | FTIR | 25.2±1.9 | |
| | | 2 | 8 | 22 | - | 23.8±0.6 | FTIR | | |
| | | 3 | 13 | 16 | - | 27.5±0.2 | SPME/GC-TOFMS[e] | | 14 |
| | Cl[a] + NO | 1 | 11 | 23 | 30 | 30.4±0.9 | FTIR | 29.5±0.7 | |
| | | 2 | 5 | 25 | 19 | 30.0±0.6 | FTIR | | |
| | | 3 | 7 | 13 | 12 | 31.6±1.3 | SPME/GC-TOFMS[e] | | |
| | OH[b] | 1 | 7 | 36 | 23 | 35.1±1.3 | FTIR | | |
| | | 2 | 13 | 31 | 29 | 38.2±1.5 | FTIR | | |
| | | 3 | 11 | 28 | 28.5 | 47.8±0.4 | FTIR | 40.2±5.4 | 53 |
| | | 4 | 6 | 19 | 12 | 39.8±0.9 | SPME/GC-TOFMS[e] | | |
| | NO$_3$[c] | 1 | 3 | 6 | - | 56.8±11.4 | SPME/GC-TOFMS[e] | | |
| | | 2 | 6 | 34 | - | 88.3±7.0 | SPME/GC-TOFMS[e] | 58.0±23.5 | 75 |
| | | 3 | 4 | 30 | - | 77.1±4.6 | SPME/GC-TOFMS[e] | | |
| | | 4 | 4 | 21 | - | 34.6±0.5 | FTIR | | |
| | | 5 | 7 | 10 | - | 33.4±0.6 | FTIR | | |

[a] Rate coefficient k (in cm$^3$ molecule$^{-1}$ s$^{-1}$ unit) used to correct the concentration of E-4-methylcyclohexanone by
loss with the reaction of Cl atoms was of 11.2 ×10$^{-11}$ (data of 2-methylcyclohexanone and Cl atoms (Herath et al.,
2018)). Photolysis rate constant estimated for E-4-methylcyclohexanone under our experimental conditions. k$_p$ =5
× 10$^{-5}$ s$^{-1}$
[b] Rate coefficient k (in cm$^3$ molecule$^{-1}$ s$^{-1}$ unit) used to correct the concentration of E-4-methylcyclohexanone by
loss with the reaction of OH radical was of 13.7 × 10$^{-12}$ (estimated using AOPWIN, v1.92). Photolysis rate constant
estimated for E-4-methylcyclohexanone under our experimental conditions. k$_p$ =5 × 10$^{-5}$ s$^{-1}$
[c] Rate coefficient k (in cm$^3$ molecule$^{-1}$ s$^{-1}$ unit) used to correct the concentration of E-4-methylcyclohexanone by
loss with the reaction of NO$_3$ radical was of 2.28 × 10$^{-16}$ (estimated using SAR method, Kerducci et al., 2014)
[d] Indicated errors are the associated error to the slope of plots obtained in the least square analysis
[e] Experiment using a Teflon gas Bag of 150 or 500 L
[f] Standard deviations 1σ





**Table 4.** Experimental conditions and yields 3,3-dimethylbutanal for the reaction of 3,3DM1ButOH
with atmospheric oxidants.

| MSA | Oxidant | Exp | [MSA] (ppm) | [Precursor] (ppm) | [NO] (ppm) | Carbonyl compound [f]Yield (%) | Technique | Average[g] (%) | SAR Yield (%) |
|---|---|---|---|---|---|---|---|---|---|
| | Cl[a] | 1 | 11 | 24 | - | 40.3±0.2 | FTIR | | |
| | | | | | | 41.8±4.6 | SPME/GC-TOFMS[d] | 39.4±15.0 | |
| | | 2 | 2.6 | 8 | - | 19.6±0.5 | SPME/GC-TOFMS[e] | | |
| | | 3 | 6 | 25 | - | 55.9±1.7 | FTIR | | |
| | Cl[a] + NO | 1 | 10 | 21 | 21 | 61.6±3.4 | FTIR | | 40 |
| | | | | | | 34.7±4.4 | SPME/GC-TOFMS[d] | 43.3±17.7 | |
| 3,3DM1ButOH | | 2 | 4 | 9 | 8 | 23.0±4.2 | SPME/GC-TOFMS[e] | | |
| | | 3 | 10 | 25 | 25 | 48.8±0.6 | FTIR | | |
| | OH[b] | 1 | 10 | 60 | 36 | 82.1±4.2 | FTIR | | 66 |
| | | | | | | 40.8±2.7 | SPME/GC-TOFMS[d] | 62.2±15.0 | |
| | | 2 | 7 | 35 | 57 | 67.4±1.4 | FTIR | | |
| | | 3 | 11 | 28 | 55 | 61.9±0.9 | FTIR | | |
| | | 4 | 11 | 29 | 30 | 59.1±3.8 | FTIR | | |
| | NO₃[c] | 1 | 11 | 36 | - | 29.2±0.5 | FTIR | | 85 |
| | | | | | | 53.9[h] | SPME/GC-TOFMS[d] | 36.2±14.6 | |
| | | 2 | 11 | 32 | - | 26.5±1.6 | FTIR | | |

[a] Rate coefficient k (in $cm^3$ molecule$^{-1}$ s$^{-1}$ unit) used to correct the concentration of 3,3-dimethylbutanal by loss
with the reaction of Cl atoms was of $1.7 \times 10^{-10}$ (data of iso-Butyraldehyde and Cl atoms (Thevenet et al., 2000)).
Photolysis rate constant estimated for 3,3-dimethylbutanal under our experimental conditions. $k_p = 1 \times 10^{-4}$ s$^{-1}$
[b] Rate coefficient k (in $cm^3$ molecule$^{-1}$ s$^{-1}$ unit) used to correct the concentration of 3,3-dimethylbutanal by loss
with the reaction of OH radical was of $2.73 \times 10^{-11}$ (Aschmann et al., 2010). Photolysis rate constant estimated for
3,3-dimethylbutanal under our experimental conditions. $k_p = 1 \times 10^{-4}$ s$^{-1}$
[c] Rate coefficient k (in $cm^3$ molecule$^{-1}$ s$^{-1}$ unit) used to correct the concentration of 3,3-dimethylbutanal by loss
with the reaction of NO₃ radical was of $1.27 \times 10^{-14}$ (D'Anna, 2001).
[d] Experiment using a FTIR Gas Cell of 50 L
[e] Experiment using a Teflon gas Bag of 150 or 500 L
[f] Indicated errors are the associated error to the slope of plots obtained in the least square analysis
[g] Standard deviations 1σ
[h] Yield estimated using only one data




**Table 5.** Experimental conditions and yields 3,3-dimethyl-2-butanone for the reaction of
3,3DM2ButOH with atmospheric oxidants

| MSA | Oxidant | Exp | [MSA] (ppm) | [Precursor] (ppm) | [NO] (ppm) | Carbonyl compound [j]Yield (%) | Technique | Average[k] (%) | SAR Yield (%) |
|---|---|---|---|---|---|---|---|---|---|
| | Cl[a] | 1 | 14 | 31 | - | 42.8±0.7 | FTIR | | |
| | | | | | | 45.2±1.1 | SPME/GC-TOFMS[d] | 43.2±1.8 | |
| | | 2 | 2.3 | 8 | - | 41.7±3.2 | SPME/GC-TOFMS[e] | | |
| | Cl[a] + NO | 1 | 14 | 28 | 20 | 36.7±5.0 | FTIR | | 40 |
| | | | | | | 49.6±4.5 | SPME/GC-TOFMS[d] | 44.2±7.4 | |
| 3,3DM2ButOH | | 2 | 3 | 7 | 6 | 39.0±6.1 | SPME/GC-TOFMS[e] | | |
| | | 3 | 8 | 28 | 27 | 51.5±3.9 | FTIR | | |
| | OH[b] | 1 | 8 | 55 | 42 | 82.8±3.1 | FTIR | | 91 |
| | | | | | | 71.2±2.6 | SPME/GC-TOFMS[d] | 80.7±6.5 | |
| | | 2 | 5 | 66 | 36 | 85.4±5.8 | FTIR | | |
| | | 3 | 11 | 28 | 29 | 83.6±3.0 | FTIR | | |
| | NO₃[c] | 1 | 12 | 30 | - | 66.7±2.05 | FTIR | | 99 |
| | | | | | | 45.9±1.6 | SPME/GC-TOFMS[d] | 58.0±10.9 | |
| | | 2 | 9 | 30 | - | 61.5±1.4 | FTIR | | |

[a] Rate coefficient k (in $cm^3$ molecule$^{-1}$ s$^{-1}$ unit) used to correct the concentration of 3,3-dimethyl-2-butanone by
loss with the reaction of Cl atoms was of $4.8 \times 10^{-11}$ (Farrugia et al., 2015)). Photolysis rate constant estimated for
3,3-dimethyl-2-butanone under our experimental conditions. $k_p = 7 \times 10^{-5}$ s$^{-1}$
[b] Rate coefficient k (in $cm^3$ molecule$^{-1}$ s$^{-1}$ unit) used to correct the concentration of 3,3-dimethyl-2-butanone by
loss with the reaction of OH radical was of $1.21 \times 10^{-12}$ (Wallington and Kurylo., 1987). Photolysis rate constant
estimated for 3,3-dimethyl-2-butanone under our experimental conditions. $k_p = 7 \times 10^{-5}$ s$^{-1}$
[c] No corrected
[d] Experiment using a FTIR Gas Cell of 50 L
[e] Experiment using a Teflon gas Bag of 150 or 500 L
[f] Standard deviations 1σ
[g] Indicated errors are the associated error to the slope of plots obtained in the least square analysis



**Table 6:** Summary of yields (%) of reaction products identified in the reaction of MSA with
atmospheric oxidants and the total carbon balance.

| Product | MSA | | | |
|---|---|---|---|---|
| | **4MCHexOH** | | | |
| | **Cl** | **Cl + NO** | **OH** | **NO$_3$** |
| **E-4-methylcyclohexanone[1]** | 25.2 ± 1.9 | 29.5 ± 0.7 | 40.2 ± 5.4 | 58.0 ± 23.5 |
| **HCOH[2]** | 9 | - | - | - |
| **Nitrated compounds** | - | 20 | 10 | 60 |
| **Total Carbon** | 34 | 50 | 50 | ~100 |
| | **3,3DM1ButOH** | | | |
| | **Cl** | **Cl + NO** | **OH** | **NO$_3$** |
| **3,3-dimethylbutanal[1]** | 39.4 ± 15.0 | 43.3 ± 17.7 | 62.2 ± 15.0 | 36.2 ± 14.6 |
| **HCOH[2]** | 10 | 22 | - | - |
| **2,2-dimethylpropanal[2]** | 22 | 8 | 23 | - |
| **Acetone[2]** | 5 | 17 | - | - |
| **Nitrated compounds** | - | 40[4] | 35[5] | 200[6] |
| **Total Carbon[3]** | 67 | 68 | 85 | 36 |
| | **3,3DM2ButOH** | | | |
| | **Cl** | **Cl + NO** | **OH** | **NO$_3$** |
| **3,3-dimethyl-2-butanone[1]** | 43.2 ± 1.8 | 44.2 ± 7.4 | 80.7 ± 6.5 | 58.0 ± 10.9 |
| **HCOH[2]** | 10 | 64 | - | - |
| **2,2-dimthylpropanal[2]** | 14 | 10 | 14 | - |
| **Acetone[2]** | 3 | 58 | - | - |
| **Acetaldehyde[2]** | - | 17 | - | - |
| **Nitrated compounds** | - | 30 | 20 | 120 |
| **Total Carbon[3]** | ~60 | 112 | 94 | 58 |

[1]Average Tables 3-5; [2]Yield obtained in earlier step of reaction.; [3]Without accounting nitrated compounds;
[4]From analysis of the experiment number 3 for the reaction of Cl + NO; [5]From average of experiments number
2, 3 and 4 for the reaction with OH; [6]From analysis of the experiments number 1 and 2 for reaction with NO$_3$.



**Table 7.** Lifetimes of 4MCHexOH, 3,3DM1ButOH and 3,3DM2ButOH.

|  | $\tau_{OH}$ (days) | $\tau_{Cl}$ [a](days) | $\tau_{Cl}$ [b](days) | $\tau_{NO3}$ (days) | $\tau_{wet}$ (years) | $\tau_{global}$[a](days) |
|---|---|---|---|---|---|---|
| **4MCHexOH** | 0.61 | 31.20 | 0.24 | 7.93 | ~2.1 | 0.55 |
| **3,3DM1ButOH** | 2.08 | 42.87 | 0.33 | 13[c] | ~15.8 | 1.72 |
| **3,3DM2ButOH** | 1.10 | 95.65 | 0.74 | 6.73[c] | 11.3 | 0.94 |

[a]Determined with the 24 hours average of chlorine atoms.
[b]Determined with the peak concentration of chlorine atoms.
[c]Determined using the rate coefficient obtained by Moreno A. et al., 2014.
A)

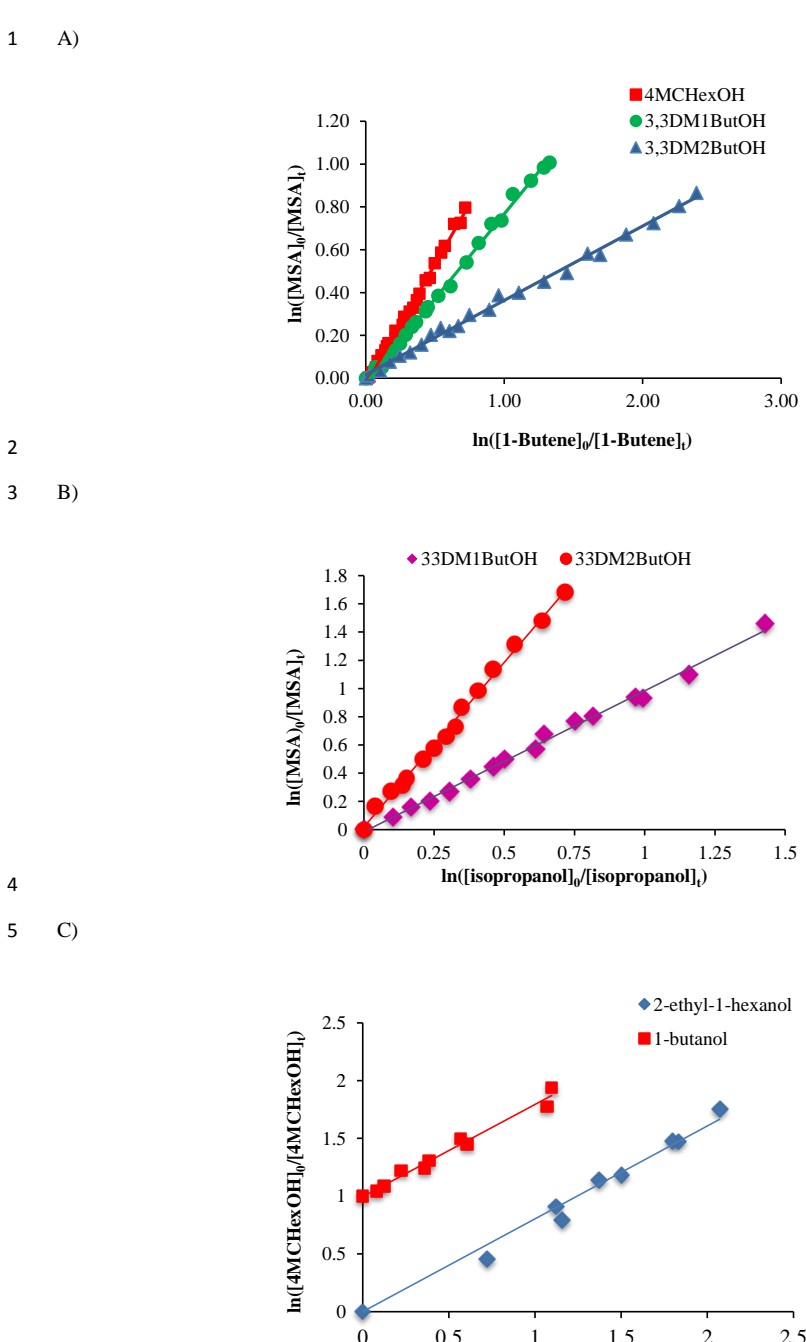

B)
C)
Fig.1: Relative rate plots for the reaction of (A) MSA with chlorine atoms employing 1-butene as a reference
compound (B) 3,3-Dimethylbutanols and OH radical with isopropanol as a reference compound and (C)
4MCHexOH and NO$_3$ with two reference compounds. Data for 1-butanol have been vertically displaced for clarity.

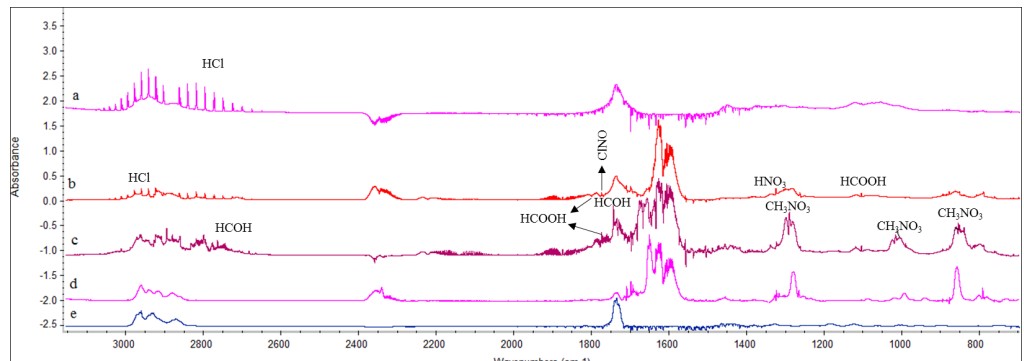

Fig. 2: Product spectra for reaction of 4MCHexOH with: (a) chlorine atoms at 10 min (x 2 to clarify), (b) chlorine
atoms and NOx at 7 min. (c) OH at 40 min and (d) NO₃ at 32 min. (e) Spectrum of 4-methylcyclohexanona
commercial sample.

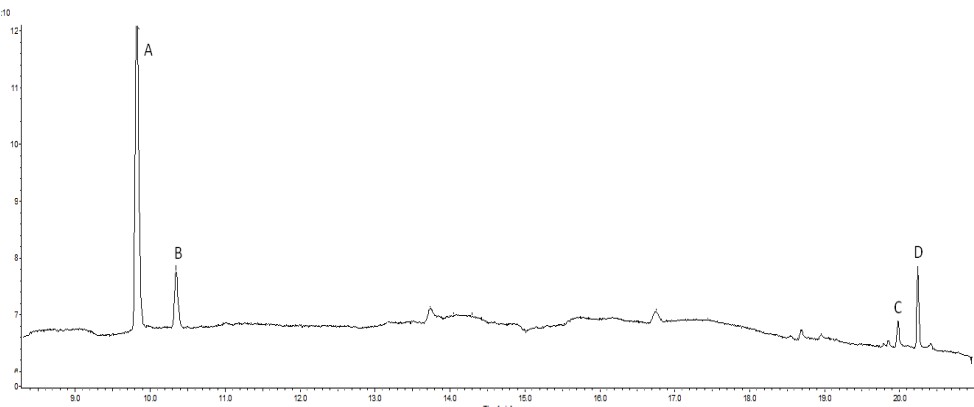

2
3  Fig. 3: SPME/GC-TOFMS chromatogram for the reaction of 4MCHexOH with chlorine atoms after 15 minutes
4  of reaction. Peak (A) 4MCHexOH. Peak (B) E-4-methylcyclohexanone.



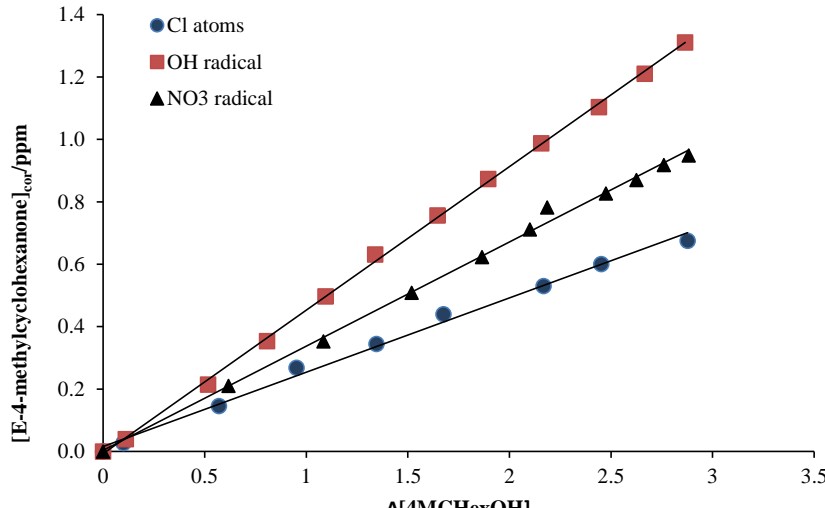

4  Fig. 4: Plots of corrected concentrations of E-4-methylcyclohexanone against 4MCHexOH consumed for Cl atoms
5  (in absence of NOx) and OH and NO$_3$ radical reactions.





Fig. 5: Reaction mechanism for the degradation of 4MCHexOH with X (Cl atom, OH and NO$_3$ radicals). (A)
Mechanism for the formation of carbonyl compounds, (B) Mechanism for the formation of nitrated compounds.
Compounds marked with solid line are positively identified. Compounds marked with shaded lines are not
positively identified.



A)

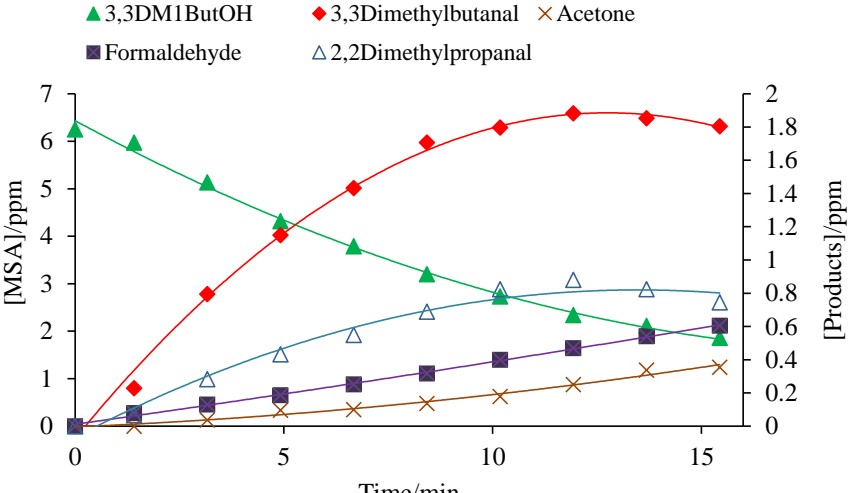

B)

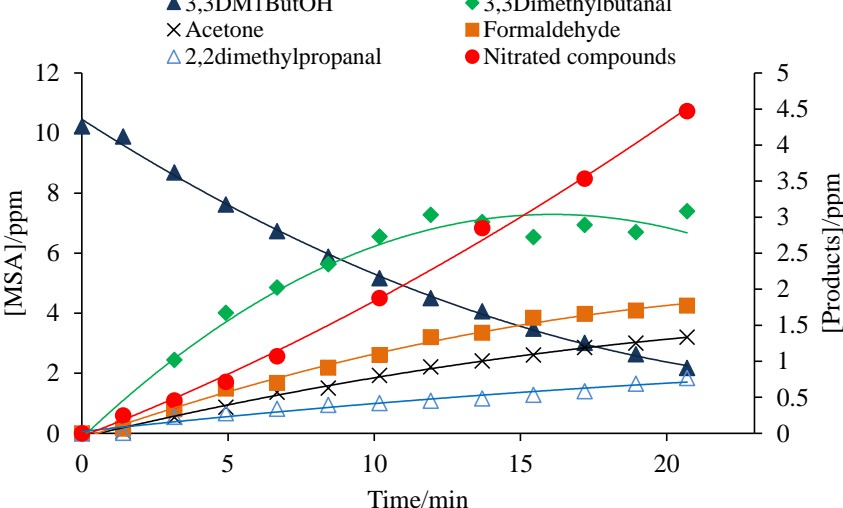

Fig. 6: Concentration-time profiles of MSA and Products for 3,3DM1ButOH with Cl atoms in absence (A) and
presence of NO (B).





Fig. 7: Reaction mechanism for the degradation of 3,3DM1ButOH with X (Cl atom, OH and NO₃ radical).
Mechanism for the formation of carbonyl compounds. Compounds marked with solid line are positively identified.
Compounds marked with shaded lines are not positively identified.





Fig. 8: Reaction mechanism for the degradation of 3,3DM2ButOH with X (Cl atom, OH and $NO_3$ radical).
Mechanism to form carbonyl compounds. Compounds marked with solid line are positively identified. Compounds
marked with shaded lines are not positively identified.
