# Peer review of "Atmospheric fate of a series of saturated alcohols: kinetic and"

_Atmospheric Chemistry and Physics, 2019_

## Referee Comment (RC1) · Anonymous Referee #1 · 19 Sep 2019

Colmenar et al. have presented in this paper an extensive study of the atmospheric chemistry of some long-chain saturated alcohols. The manuscript contains kinetic studies involving relative rate method as well as reaction product analysis for reaction with the main atmospheric oxidants. These long-chain alcohols might have potential future use in biofuels and therefore it is essential to understand the atmospheric fate of these chemicals in advance. The material of this manuscript is relevant for publication in ACP although there are scopes for improvement in terms of presentation of data and explanation of results in certain areas of the current version of the manuscript. The quality of some data is questionable and there are several typographical errors. Therefore, I recommend publishing this paper in ACP after revision considering the following issues

listed below.

Specific comments:

(i) Sec. 2.1 Kinetic experiments: The description of the experimental details for relative rate kinetic measurements involving FTIR is inadequate and some points are not clear. Is it an in situ or an ex-situ experiment? Is the White optics located inside the reaction chamber? If White absorption cell is a different cell then was there a facility for circulation of reaction mixture between the reaction chamber and the absorption cell? Are the actinic lamps located inside or outside of the reactor? I would recommend providing a schematic diagram of the whole set-up which will clarify all these issues. This would be extremely helpful for the readers to visualize and understand the whole setup.

(ii) Sec 3.1 Kinetic study: All the terms used in Table 1 should be described in this section (P 6, L 17, after the introduction of Table 1).

(iii) P 7, L 13: "the factor of hydroxyl. . .." – define this factor.

(iv) P 7, L 23 – 27: this portion is not clear. Please explain.

(v) Sec 3.1.1 Estimation of rate coefficients: The title for this section should be modified. The method used for the estimation of rate coefficient should be mentioned in the title.

(vi) P 7, L 39: When you first introduce SAR, write its full form. Also, since a lot of discussions has been made on SAR, it would be helpful to briefly describe the basics of SAR method in this section.

(vii) P 9, L 27-28: All the IR bands mentioned here are not labeled in Fig. S2. Also, the font size for the labels is too small.

(viii) P 10, L 8: "It should be noted that these data should be taken with caution, since they could imply many sources of error" – Please discuss all possible sources of error.

(ix) P 10, L 21-24: The large difference between the yields of E-4-methylcyclohexanone obtained using the SPME/GC-TOFMS and FTIR is surprising. The authors argued that
the difference in reactor volume could be the reason behind. This is not clear to me. Please explain in details.

(x) Table 1: The terms used in the table are not described either in the main text or in the legend of the table. What are the quantities listed in column 4 and 5? The values listed in Column 4 appear to be average of the values presented in Column 3, yet the same notation for the two columns was used. The uncertainties for some values are extremely high (sometimes close to 50 % !!) which is unacceptable. A detailed discussion on the possible sources and high values of the uncertainties should be presented in the text.

(xi) Table 6: Total C balance for some reactions (particularly for Cl reactions) is extremely low. Please explain.

Technical issues:

The language in some parts of the manuscript could be improved. I've noticed some typos and other technical issues throughout the manuscript which are listed below. I have not thoroughly checked for technical issues in supplementary material and I'd request the authors to review this section again.

(i) Title: Some words of the title are written in title case while other words are not. Consistency should be maintained.

(ii) Abstract: P1 L13 – is the full stop at the end of this line valid? It looks like the sentence is continuing in the next line. P1, L20: when you first introduce HCOH, write its full chemical name.

(iii) P2, L25: Change "Others" to "Other".

(iv) P3, L23: the rate coefficient for the reaction with MSA is termed as kS here while in equation (1) it is termed as kMSA. Please, correct. Also, define kS and kR here.

(v) P4, L5: equation (1) is written in Roman (I). Please change.

(vi) P4, L5: 1st and 2nd brackets are wrongly placed for both the terms.

(vii) In many places, hydrogen "subtraction" is written instead of "abstraction".

(viii) P6, L30: add "for MSA" after "... NO3 are higher".

(ix) P7, L2: write the full word "molecule", not the abbreviation "molec" in the unit.

(x) P7, L10: change "not" to "no".

(xi) P8, L1: "develop" can be changed to "developed".

(xii) P8, L19-20: check Units.

(xiii) P8, L23-24: These values could be included in Table 2.

(xiv) P8, L36: "Bands that are due. . .." – sentence is incomplete.

(xv) P9, L15: "of Fig. 3" can be changed to "in Fig. 3".

(xvi) P11, L4 (and in other places): "stablish" can be replaced by "establish".

(xvii) P14, L22: change "MSA have not a" to "MSA do not have a".

(xviii) Table 2: "Ratio" – "R" is capital in one place and small in the other two places.

(xviv) Fig.1 B): Left axis – correct problem with 1st and 3rd brackets.

(xvv) Fig2. Title: change "4-methylcyclohexanona" to "4-methylcyclohexanone".

(xvvi) Fig3. Picture quality is poor. Axis fonts are not readable.

(xvvii) Fig.4: the unit of x-axis missing. Describe the terms used in both the axis.

(xviii) Fig7 & 8: the dot sign of radical is missing in some places.

---

## Author Comment (AC1) · 26 Sep 2019

RC: Colmenar et al. have presented in this paper an extensive study of the atmospheric chemistry of some long-chain saturated alcohols. The manuscript contains kinetic studies involving relative rate method as well as reaction product analysis for reaction with the main atmospheric oxidants. These long-chain alcohols might have potential future use in biofuels and therefore it is essential to understand the atmospheric fate of these chemicals in advance. The material of this manuscript is relevant for publication in ACP although there are scopes for improvement in terms of presentation of data and explanation of results in certain areas of the current version of the manuscript. The

quality of some data is questionable and there are several typographical errors. There-fore, I recommend publishing this paper in ACP after revision considering the following issues listed below. AC: We thank the referee for the interest shown on our work and the comments and suggestions. Specific comments:

RC: (i) Sec. 2.1 Kinetic experiments: The description of the experimental details for relative kinetic measurements involving FTIR is inadequate and some points are not clear. Is it an in situ or an ex-situ experiment? Is the White optics located inside the reaction chamber? If White absorption cell is a different cell then was there a facility for circulation of reaction mixture between the reaction chamber and the absorption cell? Are the actinic lamps located inside or outside of the reactor? I would recommend pro-viding a schematic diagram of the whole set-up which will clarify all these issues. This would be extremely helpful for the readers to visualize and understand the whole setup. AC: Due to extensive number of results presented in this article, the authors have con-sidered to omit certain aspects related to the description of the experimental system and procedure, since all this information is widely described in previous works of our research group (Tapia et al 2011. https://doi.org/10.5194/acp-11-3227-2011; Martin et al. 2013. https://doi.org/10.1016/j.atmosenv.2013.01.041 ). We suggest consulting these references for more information. However, and according with the your comment we have decided to extent the description of the experimental system in the manuscript AC: In the case of the methods of estimation rate coefficients. A brief explanation of the SAR method together with the calculations developed to estimate rate coefficients will be included in the supplementary material.

RC:(ii) Kinetic study: All the terms used in Table 1 should be described in this section (P 6, L 17, after the introduction of Table 1). AC: Table 1 has been modified. See comment to the question (x). In addition, the following description that shows how errors have been calculated will be included in the main text. "The ratios of the rate coefficients, kMSA/kR, the absolute rate coefficients and the weighted average are shown in Table 1. The error of kMSA/kR are given by 2 times the statistical deviation calculated from

the least-square fit of the plot of Eq. (1). The uncertainties for rate coefficients of MSA (sigmakMSA) were calculated from the uncertainty of slope of plots (ïĄşslope) and the uncertainty of the reference (sigmakR) by using the propagation of uncertainties. The average value of the rate coefficient obtained with different reference compounds and its associated error were obtained by weighted average."

RC: (iii) P 7, L 13: "the factor of hydroxyl: : :." – define this factor. AC: P7 the lines 12 and 13 where appears "the factor of hydroxyl: : :." have been rewritten for more clarity. "SAR method has quantified this effect for each functional group of an organic compound establishing a series of factor of reactivity (F(X)) (See A1 supplementary material). In the case of.."

RC:(iv) P 7, L 23 – 27: this portion is not clear. Please explain. AC: This paragraph has been rewritten for more clarity. "As can be seen in Table S1, in the case of the Cl atoms reactions, the rate coefficients for primary alcohols (1-propanol, 1-butanol, 1-pentanol, 3-methyl-1-butanol and 3,3-dimethyl-1-butanol), are higher than the ones of the secondary alcohols (2-propanol, 2-butanol, 2-pentanol, 3-methyl-2-butanol and 3,3-dimethyl-2-butanol). This fact could be due to the more quantity of hydrogens activated in alfa position, while in the case of OH and NO3 radicals seems to be more important the formation of the most stable radical than the number of hydrogen in alfa position."

RC: (v) Sec 3.1.1 Estimation of rate coefficients: The title for this section should be modified. The method used for the estimation of rate coefficient should be mentioned in the title. AC: We have considered that it should be a generic title and not particularize, due to the fact that the estimation has been done using two different methods.

RC:(vi) P 7, L 39: When you first introduce SAR, write its full form. Also, since a lot of discussions has been made on SAR, it would be helpful to briefly describe the basics of SAR method in this section. AC: It is true that the first time SAR appears, it must be indicated to which the acronyms correspond. This will be corrected. An explication will be made in supplementary information (S1) in order to not do more extensive the

manuscript.

RC: (vii) P 9, L 27-28: All the IR bands mentioned here are not labelled in Fig. S2. Also, the font size for the labels is too small. AC: All IR bands mentioned in the main text (P9, Line 27-28) are labelled in the different spectra of Fig. S2. (Fig. S3 in the next version of supplementary materials) We have found an error of IR bands, P9 line 28, appears 1260 cm-1 but must be aprox 1660 cm-1. This IR band is labelled in Fig. S2 in the green spectrum (1652.7 cm-1). The size of the labels will be increased in the next version of the manuscript.

RC:(viii) P 10, L 8: "It should be noted that these data should be taken with caution, since they could imply many sources of error" – Please discuss all possible sources of error. AC: The two experimental systems used involve different sources of error: -Errors in the process of introducing the reagents into the gas cell or Teflon bags, (by dragging the compound into a carrier gas stream). -Error in measuring the amount of sample when injected with a micro syringe. -In the case of the experiments carried out in the FTIR, the fact that the reagents and products have similar absorption bands makes the subtraction process difficult to perform. In addition small variations in the subtraction factor can have a lot of influence on the yields of the reaction products. -In the SPME-GCTOFMS system there are systematic errors in the sampling process by the operator (off-line process). Furthermore, all the compounds present in the reaction mixture (reagents and products) compete differently for adsorbing on the fiber.

RC:(ix) P 10, L 21-24: The large difference between the yields of E-4-methylcyclohexanone obtained using the SPME/GC-TOFMS and FTIR is surprising. The authors argued that the difference in reactor volume could be the reason behind. This is not clear to me. Please explain in details. AC:We consider that the difference in yields is due to the procedure used in the different reactors for the study of the reactions with the nitrate radical. In the case of the experiments in the Teflon reactor, the volume of 150 L allows us to make small additions of the precursor (N2O5) until the final concentration indicated in Table 3. Consequently, when small precursor

amounts are added, the concentration of inorganic nitrated compounds (NO3, NO2, HNO3) in the reaction mixture is smaller than doing only one addition in excess, which is what is done in the Pyrex glass reactor, where since the initial time of reaction, there are high concentrations of these nitrated compounds. In this way, the formation of nitrated organic compounds (confirmed in the FTIR experiments) is being favoured in the 50 L reactor versus the formation of 4-methylcyclohexanone. In the new version of manuscript the paragraph of P10 lines 22-26 has been modified. "...could be due to the different way to add the precursor in both reactors (small aliquots of N2O5 in the Teflon$^{®}$ gas bag of 150L versus only one large addition in the Pyrex$^{®}$ glass gas cell). This procedure causes a lower initial concentration of inorganic species nitrated (NO3, NO2, HNO3) in reactor of 150 L than in reactor of 50 L, favouring the formation of carbonyl compounds instead of nitrated organic compounds"

RC:(x) Table 1: The terms used in the table are not described either in the main text or in the legend of the table. What are the quantities listed in column 4 and 5? The values listed in Column 4 appear to be average of the values presented in Column 3, yet the same notation for the two columns was used. The uncertainties for some values are extremely high (sometimes close to 50 % !!) which is unacceptable. A detailed discussion on the possible sources and high values of the uncertainties should be presented in the text. AC:The data in table 1 have been reviewed (Table 1 attached in supplemet). Absolute constants and their errors have been recalculated. It was found that in certain cases different criteria had been applied in the process of defining errors (sigma or 2sigma) and a mistake was also found when applying the error propagation formula. Thus, table 1 has been modified. A column has been included with the data of the relative rate coefficients and their errors (2sigma, standard deviation of the linear adjustment by least squares) and we have decided to leave only one column with the average value of absolute rate coefficient, calculated using the weighted arithmetic mean. Thus it can be verified that the experimental data (kMSA/kR) do not show large deviations. The errors of the absolute rate coefficients have been obtained

taking into account the errors associated with the reference rate coefficients and the slope using the propagation of errors. For that reason, those data obtained using a reference rate coefficient with large error show larger uncertainties. On the other hand, it is important to indicate that it is usual to find similar error values in the field of gas phase radical atmospheric chemistry, especially when the method used is the relative one. https://kinetics.nist.gov/kinetics/index.jsp Likewise, all terms presented in the table have been described in the legend.

RC:(xi) Table 6: Total C balance for some reactions (particularly for Cl reactions) is extremely low. Please explain. AC:Table 6 has been revised and it has been observed that there were some erroneous data in the calculation of the carbon balance, although it is practically similar to the initial one. The very low values of the total balance of C are explained because, as observed in the residual spectra, after eliminating all known compound bands, there are still absorption bands of compounds that couldn't be identify or quantify, since they are not commercial. These compounds could correspond to the hydroxycarbonyls and dialcohols compounds shown in the different reaction schemes.

RC:Technical issues: The language in some parts of the manuscript could be improved. I've noticed some typos and other technical issues throughout the manuscript which are listed below. I have not thoroughly checked for technical issues in supplementary material and I'd request the authors to review this section again. (i) Title: Some words of the title are written in title case while other words are not. Consistency should be maintained. (ii) Abstract: P1 L13 – is the full stop at the end of this line valid? It looks like the sentence is continuing in the next line. P1, L20: when you first introduce HCOH, write its full chemical name. (iii) P2, L25: Change "Others" to "Other". (iv) P3, L23: the rate coefficient for the reaction with MSA is termed as kS here while in equation (1) it is termed as kMSA. Please, correct. Also, define kS and kR here. (v) P4, L5: equation (1) is written in Roman (I). Please change. (vi) P4, L5: 1st and 2nd brackets are wrongly placed for both the terms. (vii) In many

places, hydrogen "subtraction" is written instead of "abstraction". (viii) P6, L30: add "for MSA" after ": : : NO3 are higher". (ix) P7, L2: write the full word "molecule", not the abbreviation "molec" in the unit. (x) P7, L10: change "not" to "no". (xi) P8, L1: "develop" can be changed to "developed". (xii) P8, L19-20: check Units. (xiii) P8, L23-24: These values could be included in Table 2. (xiv) P8, L36: "Bands that are due: : :" – sentence is incomplete. (xv) P9, L15: "of Fig. 3" can be changed to "in Fig. 3". (xvi) P11, L4 (and in other places): "stablish" can be replaced by "establish". (xvii) P14, L22: change "MSA have not a" to "MSA do not have a". (xviii) Table 2: "Ratio" – "R" is capital in one place and small in the other two places. (xviv) Fig.1 B): Left axis – correct problem with 1st and 3rd brackets. (xvv) Fig2. Title: change "4-methylcyclohexanona" to "4-methylcyclohexanone". (xvvi) Fig3. Picture quality is poor. Axis fonts are not readable. (xvvii) Fig.4: the unit of x-axis missing. Describe the terms used in both the axis. (xviii) Fig7 & 8: the dot sign of radical is missing in some places. AC:All suggestions indicated in technical issues will be taken into account and modified in the manuscript.

Please also note the supplement to this comment:
https://www.atmos-chem-phys-discuss.net/acp-2019-662/acp-2019-662-AC1-supplement.pdf

**Supplement:**

**Interactive comment of Anonymous Referee #1**

Colmenar et al. have presented in this paper an extensive study of the atmospheric chemistry of some long-chain saturated alcohols. The manuscript contains kinetic studies involving relative rate method as well as reaction product analysis for reaction with the main atmospheric oxidants. These long-chain alcohols might have potential future use in biofuels and therefore it is essential to understand the atmospheric fate of these chemicals in advance. The material of this manuscript is relevant for publication in ACP although there are scopes for improvement in terms of presentation of data and explanation of results in certain areas of the current version of the manuscript. The quality of some data is questionable and there are several typographical errors. Therefore, I recommend publishing this paper in ACP after revision considering the following issues listed below.

 We thank the referee for the interest shown on our work and the comments and suggestions.

**Specific comments:**

(i) Sec. 2.1 Kinetic experiments: The description of the experimental details for relative kinetic measurements involving FTIR is inadequate and some points are not clear. Is it an in situ or an ex-situ experiment? Is the White optics located inside the reaction chamber? If White absorption cell is a different cell then was there a facility for circulation of reaction mixture between the reaction chamber and the absorption cell? Are the actinic lamps located inside or outside of the reactor? I would recommend providing a schematic diagram of the whole set-up which will clarify all these issues. This would be extremely helpful for the readers to visualize and understand the whole setup.

Due to extensive number of results presented in this article, the authors have considered to omit certain aspects related to the description of the experimental system and procedure, since all this information is widely described in previous works of our research group (Tapia et al 2011. https://doi.org/10.5194/acp-11-3227-2011; Martin et al. 2013. https://doi.org/10.1016/j.atmosenv.2013.01.041 ). We suggest consulting these references for more information. However, and according with the your comment we have decided to extent the description of the experimental system in the manuscript

"The experimental systems are described in previous works *(Tapia et al 2011, Martin et al. 2013).* and only a brief description is shown here. Kinetic measurements were performed at room temperature (~298 K) and atmospheric pressure ~ (720 Torr) by employing two experimental set-ups: 1)-A FTIR system formed by 50 L Pyrex® glass cell couple to the FTIR spectrometer as a detection technique ("on line" analysis). The FTIR spectrometer (Thermo Nicolet 5700) is equipped with a KBr beam splitter and liquid nitrogen-cooled MCT. Typically, for each spectrum, 60 interferograms were co-added over 98 s and approximately 30-40 spectra were recorded per kinetic experiment with a spectral resolution of 1 cm$^{-1}$. 2)-Teflon ® gas bags of 150 L or 500 L with Solid Phase Micro Extraction fiber (SPME) as a pre-concentration sample method, followed by analysis on a Gas Chromatography-Mass Spectrometry system with a Time of Flight analyzer (SPME/GC-TOFMS) (AccuTOF GCv, Jeol) ("off line" analysis). Samples were collected by exposing a 50/30 mm DVB/CAR/PDMS Solid Phase Micro Extraction fiber (SPME,SUPELCO) for 5 min during the reaction and then thermally desorbed for 15 min at 250 ºC in the heated GC injection port. A capillary column (30 m × 0.3 mm id × 1.0 mm film thickness, Tracsil TRB-1701, Teknokroma) was used to separate the compounds. The chromatographic conditions used for the analysis were as follows: injector, 250 ºC; interface, 250 ºC; oven initial temperature, 40 ºC for 4 min; ramp, 30 ºC min$^{-1}$ to 120 ºC, held for 6 min, second ramp, 30 ºC min$^{-1}$ to 200 ºC, held for 3 min. The reactants are injected into the reactors (gas cell and gas

bags) from a vacuum line by dragging with a stream of carrier gas used in the reaction. Inside of Pyrex® gas cell there is a multi-reflexion system with three mirrors that allows an infrared radiation path of 2.8 to 200 meters. This Pyrex® glass gas cell is known as white cell (Saturn Series Multi-Pass cells). On the inner walls of the housing there are 8 actinic lamps (Philips, TL-40W, Actinic BL, $\lambda_{max}$ = 360nm) and 4 actinic lamps and 6 lamps emitting in the UV-Vis (Philips TUV 36W 36G T8, λmax = 254 nm) for gas cell and Teflon® bags respectively. A scheme of the experimental systems is shown in Fig. S1 of supplementary material.

The kinetic experiments, for the Cl and OH reactions, were performed by FTIR system. A spectral subtraction procedure was used to derive the concentrations of reactant and reference compounds at time t=0 and time t. The reaction of $NO_3$ with 4MCHexOH was studied using a bigger reactor (Teflon ® gas bags of 150 L or 500 L) in order to minimize the wall deposition and dilution effects of the consecutive additions of $N_2O_5$. Chlorine atoms were obtained by photolysis of $Cl_2$ at a wavelength of 360 nm using 8 actinic lamps. OH radicals were produced by photolysis of Methyl nitrite, $CH_3ONO$, in the presence of NO in air. $CH_3ONO$ was synthesized in the laboratory as described elsewhere (Taylor et al., 1980).

In the case of the methods of estimation rate coefficients. A brief explanation of the SAR method together with the calculations developed to estimate rate coefficients will be included in the supplementary material.

Kinetic study: All the terms used in Table 1 should be described in this section (P 6, L 17, after the introduction of Table 1).

Table 1 has been modified. See comment to the question (x). In addition, the following description that shows how errors have been calculated will be included in the main text.

"The ratios of the rate coefficients, $k_{MSA}/k_R$, the absolute rate coefficients and the weighted average are shown in Table 1. The error of $k_{MSA}/k_R$ are given by 2 times the statistical deviation calculated from the least-square fit of the plot of Eq. (1). The uncertainties for rate coefficients of MSA ($\sigma_{kMSA}$) were calculated from the uncertainty of slope of plots ($\sigma_{slope}$) and the uncertainty of the reference ($\sigma_{kR}$) by using the propagation of uncertainties. The average value of the rate coefficient obtained with different reference compounds and its associated error were obtained by weighted average."

 (iii) P 7, L 13: "the factor of hydroxyl: : :." – define this factor.

The lines 12 and 13 where appears "the factor of hydroxyl: : :." have been rewritten for more clarity.

"SAR method has quantified this effect for each functional group of an organic compound establishing a series of factor of reactivity ($F_{(X)}$) (See A1 supplementary material). In the case of.."

 (iv) P 7, L 23 – 27: this portion is not clear. Please explain.

This paragraph has been rewritten for more clarity.

"As can be seen in Table S1, in the case of the Cl atoms reactions, the rate coefficients for primary alcohols (1-propanol, 1-butanol, 1-pentanol, 3-methyl-1-butanol and 3,3-dimethyl-1-butanol), are higher than the ones of the secondary alcohols (2-propanol, 2-butanol, 2-pentanol, 3-methyl-2-butanol and 3,3-dimethyl-2-butanol). This fact could be due to the more quantity of hydrogens activated in $\alpha$ position, while in the case of OH and $NO_3$ radicals seems to be more important the formation of the most stable radical than the number of hydrogen in $\alpha$ position."

(v) Sec 3.1.1 Estimation of rate coefficients: The title for this section should be modified. The method used for the estimation of rate coefficient should be mentioned in the title.

We have considered that it should be a generic title and not particularize, due to the fact that the estimation has been done using two different methods.

(vi) P 7, L 39: When you first introduce SAR, write its full form. Also, since a lot of discussions has been made on SAR, it would be helpful to briefly describe the basics of SAR method in this section.

It is true that the first time SAR appears, it must be indicated to which the acronyms correspond. This will be corrected.

An explication will be made in supplementary information (S1) in order to not do more extensive the manuscript.

(vii) P 9, L 27-28: All the IR bands mentioned here are not labelled in Fig. S2. Also, the font size for the labels is too small.

All IR bands mentioned in the main text (P9, Line 27-28) are labelled in the different spectra of Fig. S2. (Fig. S3 in the next version of supplementary materials) We have found an error of IR bands, P9 line 28, appears 1260 cm$^{-1}$ but must be ~1660 cm$^{-1}$. This IR band is labelled in Fig. S2 in the green spectrum (1652.7 cm$^{-1}$).

The size of the labels will be increased in the next version of the manuscript.

(viii) P 10, L 8: "It should be noted that these data should be taken with caution, since they could imply many sources of error" – Please discuss all possible sources of error.

The two experimental systems used involve different sources of error:

Errors in the process of introducing the reagents into the gas cell or Teflon bags, (by dragging the compound into a carrier gas stream).

Error in measuring the amount of sample when injected with a micro syringe.

In the case of the experiments carried out in the FTIR, the fact that the reagents and products have similar absorption bands makes the subtraction process difficult to perform. In addition small variations in the subtraction factor can have a lot of influence on the yields of the reaction products.

In the SPME-GCTOFMS system there are systematic errors in the sampling process by the operator (off-line process). Furthermore, all the compounds present in the reaction mixture (reagents and products) compete differently for adsorbing on the fiber.

(ix) P 10, L 21-24: The large difference between the yields of E-4-methylcyclohexanone obtained using the SPME/GC-TOFMS and FTIR is surprising. The authors argued that the difference in reactor volume could be the reason behind. This is not clear to me. Please explain in details.

We consider that the difference in yields is due to the procedure used in the different reactors for the study of the reactions with the nitrate radical. In the case of the experiments in the Teflon reactor, the volume of 150 L allows us to make small additions of the precursor (N$_2$O$_5$) until the final concentration indicated in Table 3. Consequently, when small precursor amounts are added, the concentration of inorganic nitrated compounds (NO$_3$, NO$_2$, HNO$_3$) in the reaction mixture is smaller than doing only one addition in excess, which is what is done in the Pyrex glass reactor, where since the initial time of reaction, there are high concentrations of these nitrated compounds. In this way, the formation of nitrated organic compounds (confirmed in the FTIR experiments) is being favoured in the 50 L reactor versus the formation of 4-methylcyclohexanone.
In the new version of manuscript the paragraph of P10 lines 22-26 has been modified.

"…could be due to the different way to add the precursor in both reactors (small aliquots of $N_2O_5$ in the Teflon® gas bag of 150L versus only one large addition in the Pyrex® glass gas cell). This procedure causes a lower initial concentration of inorganic species nitrated ($NO_3$, $NO_2$, $HNO_3$) in reactor of 150 L than in reactor of 50 L, favouring the formation of carbonyl compounds instead of nitrated organic compounds"

(x) Table 1: The terms used in the table are not described either in the main text or in the legend of the table. What are the quantities listed in column 4 and 5? The values listed in Column 4 appear to be average of the values presented in Column 3, yet the same notation for the two columns was used. The uncertainties for some values are extremely high (sometimes close to 50 % !!) which is unacceptable. A detailed discussion on the possible sources and high values of the uncertainties should be presented in the text.

The data in table 1 have been reviewed. Absolute constants and their errors have been recalculated. It was found that in certain cases different criteria had been applied in the process of defining errors ($\sigma$ or $2\sigma$) and a mistake was also found when applying the error propagation formula. Thus, table 1 has been modified. A column has been included with the data of the relative rate coefficients and their errors ($2\sigma$, standard deviation of the linear adjustment by least squares) and we have decided to leave only one column with the average value of absolute rate coefficient, calculated using the weighted arithmetic mean.

Thus it can be verified that the experimental data ($k_{MSA}/k_R$) do not show large deviations. The errors of the absolute rate coefficients have been obtained taking into account the errors associated with the reference rate coefficients and the slope using the propagation of errors. For that reason, those data obtained using a reference rate coefficient with large error show larger uncertainties. On the other hand, it is important to indicate that it is usual to find similar error values in the field of gas phase radical atmospheric chemistry, especially when the method used is the relative one. https://kinetics.nist.gov/kinetics/index.jsp

Likewise, all terms presented in the table have been described in the legend.

**Table 1.** Rate coefficient ratios, absolute coefficients and average rate coefficients for the reactions of a series of MSA with Cl atoms and OH and $NO_3$ radicals at 298 K and ~ 720 Torr of pressure. Rate coefficients, k, in $cm^3$ $molecule^{-1}$ $s^{-1}$.

| Reaction | Reference | $(k_{MSA}/k_R)\pm2\sigma$ | $(k_{MSA}\pm2\sigma)^a/10^{-10}$ | $(\bar{k}_{MSA}\pm2\sigma)^b/10^{-10}$ |
|---|---|---|---|---|
| 3,3DM1ButOH + Cl | 1-butene | $0.85 \pm 0.03$ | $2.89 \pm 0.42$ | $2.69 \pm 0.16$ |
| | | $0.79 \pm 0.02$ | $2.68 \pm 0.38$ | |
| | | $0.76 \pm 0.02$ | $2.58 \pm 0.37$ | |
| | Propene | $1.18 \pm 0.02$ | $2.63 \pm 0.37$ | |
| | | $1.21 \pm 0.03$ | $2.70 \pm 0.38$ | |
| | | $1.22 \pm 0.03$ | $2.71 \pm 0.38$ | |
| 3,3DM2ButOH + Cl | 1-butene | $0.42 \pm 0.01$ | $1.42 \pm 0.21$ | $1.21 \pm 0.07$ |
| | | $0.35 \pm 0.01$ | $1.17 \pm 0.17$ | |
| | | $0.41 \pm 0.01$ | $1.38 \pm 0.20$ | |
| | Propene | $0.48 \pm 0.01$ | $1.08 \pm 0.15$ | |
| | | $0.50 \pm 0.02$ | $1.12 \pm 0.16$ | |
| | | $0.56 \pm 0.03$ | $1.26 \pm 0.19$ | |
| 4MCHexOH + Cl | 2-methylpropene | $1.08 \pm 0.03$ | $3.69 \pm 0.32$ | $3.70 \pm 0.16$ |
| | | $1.16 \pm 0.02$ | $3.95 \pm 0.33$ | |
| | | $0.98 \pm 0.05$ | $3.35 \pm 0.32$ | |
| | 1-butene | $1.14 \pm 0.03$ | $3.86 \pm 0.56$ | |
| | | $1.12 \pm 0.03$ | $3.78 \pm 0.55$ | |
| | | $1.15 \pm 0.04$ | $3.90 \pm 0.57$ | |

| Reaction | Reference | $(k_{MSA}/k_R)\pm2\sigma$ | $(k_{MSA}\pm2\sigma)/10^{-12}$ | $\bar{k}_{MSA}\pm2\sigma/10^{-12}$ |
|---|---|---|---|---|
| 3,3DM1BuOH + OH | Isopropanol | $1.00 \pm 0.04$ | $5.09 \pm 0.20$ | $5.33 \pm 0.16$ |
| | | $1.13 \pm 0.09$ | $5.78 \pm 0.47$ | |
| | | $1.12 \pm 0.08$ | $5.72 \pm 0.40$ | |
| | 2-methyl-2-Butanol | $1.60 \pm 0.09$ | $5.78 \pm 1.01$ | |
| | | $1.57 \pm 0.08$ | $5.65 \pm 1.00$ | |
| | | $1.61 \pm 0.09$ | $5.79 \pm 1.02$ | |
| 3,3DM2BuOH + OH | Isopropanol | $2.33 \pm 0.09$ | $11.90 \pm 0.48$ | $10.50 \pm 0.25$ |
| | | $2.05 \pm 0.08$ | $10.50 \pm 0.45$ | |
| | | $1.95 \pm 0.08$ | $9.95 \pm 0.43$ | |
| | 2-methyl-2-butanol | $2.39 \pm 0.09$ | $8.61 \pm 1.50$ | |
| | | $2.92 \pm 0.09$ | $10.50 \pm1.78$ | |
| | | $2.25 \pm 0.09$ | $8.12 \pm 1.34$ | |
| 4MCHexOH + OH | Propene | $0.64 \pm 0.01$ | $17.10 \pm 2.59$ | $18.70 \pm 1.42$ |
| | | $0.76 \pm 0.03$ | $20.30 \pm 3.19$ | |
| | | $0.76 \pm 0.02$ | $20.40 \pm 3.10$ | |
| | Cyclohexene | $0.27 \pm 0.01$ | $18.20 \pm 4.55$ | |
| | | $0.27 \pm 0.01$ | $18.40 \pm 4.62$ | |
| | | $0.27 \pm 0.01$ | $18.00 \pm 4.46$ | |

| Reaction | Reference | $(k_{MSA}/k_R)\pm2\sigma$ | $(k_{NO3}\pm2\sigma)/10^{-15}$ | $\bar{k}_{NO3}\pm2\sigma/10^{-15}$ |
|---|---|---|---|---|
| 4MCHexOH + $NO_3$ | 1-butanol | $1.08 \pm 0.12$ | $3.39 \pm 1.11$ | $2.69 \pm 0.37$ |
| | | $1.81 \pm 0.15$ | $5.70 \pm 1.82$ | |
| | | $0.79 \pm 0.07$ | $2.51 \pm 0.80$ | |
| | 2-ethyl-1-hexanol | $0.71 \pm 0.10$ | $2.08 \pm 0.72$ | |
| | | $1.00 \pm 0.10$ | $2.93 \pm 0.96$ | |
| | | $0.86 \pm 0.08$ | $2.52 \pm 0.82$ | |

[a]The uncertainties for rate coefficients of MSA were calculated from the uncertainty of slope of plots and the uncertainty of the reference by using the propagation of uncertainties. [b]Weighted average according to the precision of the measurement ($w=1/\sigma^2$).

(xi) Table 6: Total C balance for some reactions (particularly for Cl reactions) is extremely low. Please explain.

Table 6 has been revised and it has been observed that there were some erroneous data in the calculation of the carbon balance, although it is practically similar to the initial one.
The very low values of the total balance of C are explained because, as observed in the residual spectra, after eliminating all known compound bands, there are still absorption bands of compounds that couldn't be identify or quantify, since they are not commercial. These compounds could correspond to the hydroxycarbonyls and dialcohols compounds shown in the different reaction schemes.

**Technical issues:**
**The language in some parts of the manuscript could be improved. I've noticed some typos and other technical issues throughout the manuscript which are listed below. I have not thoroughly checked for technical issues in supplementary material and I'd request the authors to review this section again.**
(i) Title: Some words of the title are written in title case while other words are not. Consistency should be maintained.
(ii) Abstract: P1 L13 – is the full stop at the end of this line valid? It looks like the sentence is continuing in the next line. P1, L20: when you first introduce HCOH, write its full chemical name.
(iii) P2, L25: Change "Others" to "Other".
(iv) P3, L23: the rate coefficient for the reaction with MSA is termed as kS here while in equation (1) it is termed as kMSA. Please, correct. Also, define kS and kR here.
(v)  P4, L5: equation (1) is written in Roman (I). Please change.

(vi) P4, L5: 1st and 2nd brackets are wrongly placed for both the terms.

(vii) In many places, hydrogen "subtraction" is written instead of "abstraction".
(viii) P6, L30: add "for MSA" after ": : : NO3 are higher".
(ix) P7, L2: write the full word "molecule", not the abbreviation "molec" in the unit.
(x) P7, L10: change "not" to "no".
(xi) P8, L1: "develop" can be changed to "developed".
(xii) P8, L19-20: check Units.
(xiii) P8, L23-24: These values could be included in Table 2.
(xiv) P8, L36: "Bands that are due: : :." – sentence is incomplete.
(xv) P9, L15: "of Fig. 3" can be changed to "in Fig. 3".
(xvi) P11, L4 (and in other places): "stablish" can be replaced by "establish".
(xvii) P14, L22: change "MSA have not a" to "MSA do not have a".
(xviii) Table 2: "Ratio" – "R" is capital in one place and small in the other two places.
(xviv) Fig.1 B): Left axis – correct problem with 1st and 3rd brackets.
(xvv) Fig2. Title: change "4-methylcyclohexanona" to "4-methylcyclohexanone".
(xvvi) Fig3. Picture quality is poor. Axis fonts are not readable.
(xvvii) Fig.4: the unit of x-axis missing. Describe the terms used in both the axis.
(xviii) Fig7 & 8: the dot sign of radical is missing in some places.

All suggestions indicated in technical issues will be taken into account and modified in the manuscript.

---

## Referee Comment (RC2) · Anonymous Referee #2 · 7 Oct 2019

**Manuscript Number: ACP-2019-662**

**Title: Atmospheric fate of a series of Methyl Saturated Alcohols (MSA): Kinetic and Mechanistic study**

**General comments:** This paper reports the experimental studies on the reactions of Cl atoms, OH radicals and $NO_3$ with MSA using Relative rate method using FTIR and GC-TOFMS as analytical tools. They have carried out the product analysis at room temperature in presence of synthetic air and reported the products obtained for the title reactions.

**Recommendation: This work is good and carried out systematically but, of routine nature not suitable to ACP and can be published in more specific journals related to kinetics. However, the authors may consider the following suggestion to improve the quality of the Paper, if they wish to submit to another specific journal.**

**Major issues regarding the manuscripts:**

1. The manuscript is difficult to read and understand, confusing in many places, careful reading should be done throughout.

2. In the abstract, "$(2.70 \pm 0.55) \times 10^{-10}$ and $(5.57 \pm 0.66) \times 10^{-12}$ for reaction of 3,3-dimethyl-1-butanol with Cl and OH radical respectively and $(1.21 \pm 0.37) \times 10^{-10}$ and $(10.51 \pm 0.81) \times 10^{-12}$ for reaction of 3,3-dimethyl-2-butanol with Cl and OH radical respectively". – sentence should be rewritten.

3. In page no.2; "Therefore, previously to the massive use, it is necessary to study the reactivity of the large alcohols in atmospheric conditions, in order to establish and to evaluate their atmospheric impact". – the atmospheric conditions may vary depends upon the altitude hence temperature dependent and pressure dependent studies need to be done in order to get the complete atmospheric impact.

4. What are the limits for photolysis and wall effect limits? Is there any preliminary reaction carried out to check secondary chemistry for the title reactions? Explain.

5. Page 4, "Kinetic measurements were performed at room temperature 298 K) and atmospheric pressure ~ (720 Torr)" – Authors have stated the pressure at which the reactions were carried out is 720 Torr throughout the main text but, in the abstract it is stated as 740 Torr. It is advised to check the values.

6. $CH_3ONO$ was synthesized in the laboratory – give the procedure and specify the purity of the prepared compound with NMR, IR etc.

7. Page 5, "During the reaction process in the 50 L Pyrex® glass chamber, the identification of products was made using the FTIR analysis but, at the same time, a sample was taken and analyzed in the SPME/GC-TOFMS system". -  is quite confusing and should be rephrased for better understanding for the readers.

8. Page 5, "To obtain the yield in percentage of carbon, the yield obtained is multiplied by 100 and by the ratio of carbons between the product and the MSA from which it comes". – not clear.

9. Authors are advised to use the recommended rate coefficients for all the reference reactions for better reliability of the rate coefficients.

10. Page 7, "This behavior could be explained for the different size and electronic properties of each oxidant that make the Cl atom the most reactive (value of k in the limit of collision) but also less selective than OH and $NO_3$ radicals". – needs more explanation.

11. Page 7, "In the case of 3,3-dimethylbutanols, there is………… of the structure of the organic compound on the reactivity (SAR Method, Kwok and Atkinson, 1995)". -Rewrite the sentence.

12. Page 7, "The activating effect of the length chain in the reactivity is being more marked in the Cl reaction than in the case of OH and $NO_3$ reactions". Why? Proper explanation should be given. Sentence is very confusing.

13. Page 8, "In general, the SAR method applied to alcohols predicts better rate coefficients for Cl atoms and OH radical than for $NO_3$ radical, especially for primary alcohols". – But the value of the $k_{exp}/k_{SAR}$ for the reaction of 3,3DM1ButOH with $NO_3$ is found out to be 3.29. Please give the explanation for this discrepancy.

14. Page 8, "….and in some cases due to heterogeneous reactions with the walls of the gas cell". – contradicting statement - check the experimental method given!

15. Page 10, why no exploration on $OH + NO$ and $NO_2 + NO_3$??

16. "The kinetic and product study confirms that the atmospheric degradation mechanism 1 for methyl saturated alcohols and possibly for the rest of unstudied saturated alcohols, proceeds mainly by abstraction of the hydrogen atom bonded to carbon instead hydrogen atoms bonded to oxygen atom of the alcohol group". – This is a known fact and should be removed from the conclusion.

17. Main text and Table 2 values – The given reasons are different. Please clarify.

18. Since 2-butanol is not a suitable reference authors could have been chosen another reference for their studies.

19. In Table 1, it seems like authors have taken the average of deviation values obtained in individual rate coefficients (column 4). It is advised to carry out the proper analysis of the errors by standard error propagation method. (For reference see *Chem. Phys. Lett*. 2013, 590, 221-226 and *New J. Chem*. 2017, 41, 7491-7505).

20. Why the effect of the bath gas on the rate coefficients were not explored?

---

## Author Response (AR1)

**Response to the anonymous referee #1**

**Taking in account the comments of referee#2 some modifications have been realized with respect the first replies. So, the authors have been considered submit again the author comments to the referee #1**

In the following, the referee's comments (RC) are reproduced (black) along with our replies AC (blue) and changes made to the text (red) in the revised manuscript.

**comment of Anonymous Referee #1**

RC: Colmenar et al. have presented in this paper an extensive study of the atmospheric chemistry of some long-chain saturated alcohols. The manuscript contains kinetic studies involving relative rate method as well as reaction product analysis for reaction with the main atmospheric oxidants. These long-chain alcohols might have potential future use in biofuels and therefore it is essential to understand the atmospheric fate of these chemicals in advance. The material of this manuscript is relevant for publication in ACP although there are scopes for improvement in terms of presentation of data and explanation of results in certain areas of the current version of the manuscript. The quality of some data is questionable and there are several typographical errors. Therefore, I recommend publishing this paper in ACP after revision considering the following issues listed below.

AC: We thank the referee for the interest shown on our work and the comments and suggestions.

**Specific comments:**

RC: (i) Sec. 2.1 Kinetic experiments: The description of the experimental details for relative kinetic measurements involving FTIR is inadequate and some points are not clear. Is it an in situ or an ex-situ experiment? Is the White optics located inside the reaction chamber? If White absorption cell is a different cell then was there a facility for circulation of reaction mixture between the reaction chamber and the absorption cell? Are the actinic lamps located inside or outside of the reactor? I would recommend providing a schematic diagram of the whole set-up which will clarify all these issues. This would be extremely helpful for the readers to visualize and understand the whole setup.

AC: Due to extensive number of results presented in this article, the authors have considered to omit certain aspects related to the description of the experimental system and procedure, since all this information is widely described in previous works of our research group (Tapia et al 2011. https://doi.org/10.5194/acp-11-3227-2011; Martin et al. 2013. https://doi.org/10.1016/j.atmosenv.2013.01.041 ). We suggest consulting these references for more information. However, and according to your comment we have decided to extent the description of the experimental system in the manuscript

"The experimental systems are described in previous works (Tapia et al 2011, Martin et al. 2013) and only a brief description is shown here. Kinetic measurements were performed at room temperature (~ 298 K) and atmospheric pressure (720 $\pm$ 20 Torr) by employing two separated experimental set-ups: 1) -A FTIR system formed by 50 L Pyrex® glass reactor couple to the Fourier Transform Infrared Radiation spectrometer as a detection technique ("on line" analysis). Inside of Pyrex® glass reactor there is a multi-reflexion system with three mirrors that allows an infrared radiation path of 2.8-200 meters. This reactor is known as white cell (Saturn Series Multi-Pass cell). The FTIR spectrometer (Thermo Nicolet 6700) is equipped with a KBr beam splitter and liquid nitrogen-cooled MCT. Typically, for each spectrum, 60 interferograms were co-added over 98 s and approximately 30-40 spectra were recorded per experiment with a spectral resolution of 1 cm-1. 2) -Teflon ® gas bag reactor of 500 L with Solid Phase Micro Extraction fiber (SPME) as a pre-concentration sample method, followed by analysis on a Gas Chromatography-Mass Spectrometry system with a Time of Flight analyzer (SPME/GC-TOFMS) (AccuTOF GCv, Jeol) ("off line" analysis). Samples were collected by exposing a 50/30 mm DVB/CAR/PDMS Solid Phase Micro Extraction fiber (SPME, SUPELCO) for 5 min during the reaction and then thermally desorbed for 15 min at 250 ºC in the heated GC injection port. A capillary column (30 m × 0.3 mm id × 1.0 mm film thickness, Tracsil TRB-1701, Teknokroma) was used to separate the compounds. The chromatographic conditions used for the analysis were as follows: injector, 250 ºC; interface, 250 ºC; oven initial temperature, 40 ºC for 4 min; ramp, 30 ºC min-1 to 120 ºC, held for 6 min; second ramp, 30 ºC min-1 to 200 ºC, held for 3 min.

In each independent experiment, the reactants are injected into the reactors from a vacuum line by dragging with a stream of carrier gas used in the reaction. Both reactors are inside of a metallic housing in which walls there is a rack of actinic lamps (Philips, TL-40W, Actinic BL, λmax = 360 nm). A scheme of the experimental systems is shown in Fig. S1 of supplementary material.

The kinetic experiments, for the Cl and OH reactions, were performed FTIR system. A spectral subtraction procedure was used to derive the concentrations of reactant and reference compounds at time t=0 and time t. The reaction of NO3 with 4MCHexOH was studied using a Teflon ® reactor of 500 L in order to minimize the wall deposition and dilution effects of consecutive additions of N2O5. Chlorine atoms and OH radicals were obtained by photolysis of Cl2 in N2 and methyl nitrite in the presence of NO in air. Methyl nitrite, CH3ONO, was synthesized in the laboratory as described elsewhere (Taylor et al., 1980)."

AC: In the case of the methods of estimation rate coefficients. A brief explanation of the SAR method together with the calculations developed to estimate rate coefficients have been included in the supplementary material.

RC: (ii) Kinetic study: All the terms used in Table 1 should be described in this section (P 6, L 17, after the introduction of Table 1).

AC: Table 1 has been modified. See comment to the question (x). In addition, the following description that shows how errors have been calculated have been included in the main text.

"The ratios of the rate coefficients, $k_{MSA}/k_R$, the absolute rate coefficients and the weighted average are shown in Table 1. The error of $k_{MSA}/k_R$ are given by 2 times the statistical deviation calculated from the least-square fit of the plot of Eq. (1). The uncertainties for rate coefficients of MSA ($\sigma_{kMSA}$) were calculated from the uncertainty of slope of plots ($\sigma_{slope}$) and the uncertainty of the reference ($\sigma_{kR}$) by using the propagation of uncertainties. The average value of the rate coefficient obtained with different reference compounds and its associated error were obtained by weighted average."

RC: (iii) P 7, L 13: "the factor of hydroxyl: : :." – define this factor.

AC: Taking in account your suggestion and the suggestion of referee #2, this part of the kinetic discussion has been rewritten/reorganized in order to more clarity. In the new version of manuscript, the next sentences appear:

−" The activating effect of hydroxyl group of the alcohols was quantified by different authors (Kwok and Atkinson 1995; Kerdouci et al, 2010; Calvert et al. 2011) taking into account the available kinetic data reported in bibliography, obtaining the factor of reactivity for the hydroxyl group, F(-OH)). This factor of reactivity is different for each oxidant, 1.18 for Cl reaction, 2.35 for reaction with OH (Calvert et al. 2011) and 18 for $NO_3$ reaction (Kerdouci et al., 2010). There are no data of rate coefficients for the reactions of the homologous alkanes of the MSA studied in this work with $NO_3$ radical, and therefore it is not possible to check out the effect of hydroxyl group in the reactivity of $NO_3$ reaction. However, according to the factor of reactivity obtained by Kerdouci et al. (2010) for the reactions of alcohols with $NO_3$, this effect is higher than the corresponding to Cl and OH reactions."

RC: (iv) P 7, L 23 – 27: this portion is not clear. Please explain.

AC: Taking in account your suggestions and the suggestions of referee #2, the kinetic discussion section has been rewritten in order to more clarity. In the new version of manuscript, the next sentence appear:

"In addition, as can be seen in Table S1, the position of hydroxyl group has a different effect depending on the oxidant. In the case of the Cl atom reactions, the rate coefficients for primary alcohols (1-propanol, 1-butanol, 1-pentanol, 3-methyl-1-butanol and 3,3-dimethyl-1-butanol) are higher than the ones of the secondary alcohols (2-propanol, 2-butanol, 2-pentanol, 3-methyl-2-butanol and 3,3-dimethyl-2-butanol) contrary to the OH and $NO_3$ radical reactions. This fact indicates that in the reaction of Cl atoms the formation of the most stable radical seems to have less importance in the reactivity than the number of hydrogens in $\alpha$ position available to be abstracted."

RC: (v) Sec 3.1.1 Estimation of rate coefficients: The title for this section should be modified. The method used for the estimation of rate coefficient should be mentioned in the title.

AC: We have considered that it should be a generic title and not particularize, due to the fact that the estimation has been done using two different methods.

RC:(vi) P 7, L 39: When you first introduce SAR, write its full form. Also, since a lot of discussions has been made on SAR, it would be helpful to briefly describe the basics of SAR method in this section.

AC: It is true that the first time SAR appears, it must be indicated to which the acronyms correspond. This will be corrected.

An explication will be made in supplementary information (S2) in order to not do more extensive the manuscript.

RC:(vii) P 9, L 27-28: All the IR bands mentioned here are not labelled in Fig. S2. Also, the font size for the labels is too small.

AC: All IR bands mentioned in the main text (P9, Line 27-28) are labelled in the different spectra of Fig. S2. (Fig. S3 in the new version of supplementary materials). We have found an error of IR bands, P9 line 28, appears 1260 cm$^{-1}$ but must be ~1660 cm$^{-1}$. This IR band is labelled in Fig. S2 in the green spectrum (1652.7 cm$^{-1}$).

The size of the labels will be increased in the new version of the manuscript.

RC:(viii) P 10, L 8: "It should be noted that these data should be taken with caution, since they could imply many sources of error" – Please discuss all possible sources of error.

AC: The two experimental systems used involve different sources of error:

Errors in the process of introducing the reagents into the gas cell or Teflon bags, (by dragging the compound into a carrier gas stream).

Error in measuring the amount of sample when injected with a micro syringe.

In the case of the experiments carried out in the FTIR, the fact that the reagents and products have similar absorption bands makes the subtraction process difficult to perform. In addition, small variations in the subtraction factor can have a lot of influence on the molecular yields of the reaction products.

In the SPME-GCTOFMS system there are systematic errors in the sampling process by the operator (off-line process). Furthermore, all the compounds present in the reaction mixture (reagents and products) compete differently for adsorbing on the fiber.

RC:(ix) P 10, L 21-24: The large difference between the yields of E-4-methylcyclohexanone obtained using the SPME/GC-TOFMS and FTIR is surprising. The authors argued that the difference in reactor volume could be the reason behind. This is not clear to me. Please explain in details.

AC: We consider that the difference in molecular yields is due to the procedure used in the different reactors for the study of the reactions with the nitrate radical. In the case of the experiments in the Teflon reactor, the volume of 150 L allows us to make small additions of the precursor ($N_2O_5$) until the final concentration indicated in Table 3. Consequently, when small precursor amounts are added, the concentration of inorganic nitrated compounds ($NO_3$, $NO_2$, $HNO_3$) in the reaction mixture is smaller than doing only one addition in excess, which is what is done in the Pyrex glass reactor, where since the initial time of reaction, there are high concentrations of these nitrated compounds. In this way, the formation of nitrated organic compounds (confirmed in the FTIR experiments) is being favored in the 50 L reactor versus the formation of 4-methylcyclohexanone.
In the new version of manuscript the next paragraph is included.

 "…could be due to the different way to add the precursor in both reactors (small aliquots of $N_2O_5$ in the Teflon® reactor of 150L versus only one large addition in the Pyrex® reactor). This procedure causes a lower initial concentration of nitrated inorganic species ($NO_3$, $NO_2$, $HNO_3$) in reactor of 150 L than in reactor of 50 L, favoring the formation of carbonyl compounds instead of nitrated organic compounds"

RC: (x) Table 1: The terms used in the table are not described either in the main text or in the legend of the table. What are the quantities listed in column 4 and 5? The values listed in Column 4 appear to be average of the values presented in Column 3, yet the same notation for the two columns was used. The uncertainties for some values are extremely high (sometimes close to 50 % !!) which is unacceptable. A detailed discussion on the possible sources and high values of the uncertainties should be presented in the text.

AC: The data in table 1 have been reviewed. Absolute rate coefficients and their errors have been recalculated. It was found that in certain cases different criteria had been applied in the process of defining errors ($\sigma$ or $2\sigma$) and a mistake was also found when applying the error propagation formula. Thus, table 1 has been modified. A column has been included with the data of the relative rate coefficients and their errors ($2\sigma$, standard deviation of the linear adjustment by least squares) and we have decided to leave only one column with the average value of absolute rate coefficient, calculated using the weighted arithmetic mean.
Thus it can be verified that the experimental data ($k_{MSA}/k_R$) do not show large deviations. The errors of the absolute rate coefficients have been obtained taking into account the errors associated with the reference rate coefficients and the slope using the propagation of errors.

For that reason, those data obtained using a reference rate coefficient with large error show larger uncertainties. On the other hand, it is important to indicate that it is usual to find similar error values in the field of gas phase radical atmospheric chemistry, especially when the method used is the relative one. https://kinetics.nist.gov/kinetics/index.jsp. Likewise, all terms presented in the table have been described in the legend.

**Table 1.** Rate coefficient ratios, absolute rate coefficients and average rate coefficients for the reactions of a series of MSA with Cl atoms and OH and $NO_3$ radicals at 298 K and ~720 ± 20 Torr of pressure. Rate coefficients, k, in $cm^3$ molecule$^{-1}$ s$^{-1}$.

| Reaction | Reference | $(k_{MSA}/k_R) \pm 2\sigma$ | $(k_{MSA} \pm 2\sigma)^a /10^{-10}$ | $(\bar{k}_{MSA} \pm 2\sigma)^b /10^{-10}$ |
|---|---|---|---|---|
| 3,3DM1ButOH + Cl | 1-butene | 0.85 ± 0.03 | 2.89 ± 0.42 | |
| | | 0.79 ± 0.02 | 2.68 ± 0.38 | |
| | | 0.76 ± 0.02 | 2.58 ± 0.37 | |
| | Propene | 1.18 ± 0.02 | 2.63 ± 0.37 | 2.69 ± 0.16 |
| | | 1.21 ± 0.03 | 2.70 ± 0.38 | |
| | | 1.22 ± 0.03 | 2.71 ± 0.38 | |
| 3,3DM2ButOH + Cl | 1-butene | 0.42 ± 0.01 | 1.42 ± 0.21 | |
| | | 0.35 ± 0.01 | 1.17 ± 0.17 | |
| | | 0.41 ± 0.01 | 1.38 ± 0.20 | 1.21 ± 0.07 |
| | Propene | 0.48 ± 0.01 | 1.08 ± 0.15 | |
| | | 0.50 ± 0.02 | 1.12 ± 0.16 | |
| | | 0.56 ± 0.03 | 1.26 ± 0.19 | |
| 4MCHexOH + Cl | 2-methylpropene | 1.08 ± 0.03 | 3.69 ± 0.32 | |
| | | 1.16 ± 0.02 | 3.95 ± 0.33 | |
| | | 0.98 ± 0.05 | 3.35 ± 0.32 | |
| | | 1.14 ± 0.03 | 3.86 ± 0.56 | 3.70 ± 0.16 |
| | 1-butene | 1.12 ± 0.03 | 3.78 ± 0.55 | |
| | | 1.15 ± 0.04 | 3.90 ± 0.57 | |

| Reaction | Reference | $(k_{MSA}/k_R) \pm 2\sigma$ | $(k_{MSA} \pm 2\sigma)/10^{-12}$ | $\bar{k}_{MSA} \pm 2\sigma /10^{-12}$ |
|---|---|---|---|---|
| 3,3DM1BuOH + OH | Isopropanol | 1.00 ± 0.04 | 5.09 ± 0.20 | |
| | | 1.13 ± 0.09 | 5.78 ± 0.47 | |
| | | 1.12 ± 0.08 | 5.72 ± 0.40 | 5.33 ± 0.16 |
| | 2-methyl-2-Butanol | 1.60 ± 0.09 | 5.78 ± 1.01 | |
| | | 1.57 ± 0.08 | 5.65 ± 1.00 | |
| | | 1.61 ± 0.09 | 5.79 ± 1.02 | |
| 3,3DM2BuOH + OH | Isopropanol | 2.33 ± 0.09 | 11.90 ± 0.48 | |
| | | 2.05 ± 0.08 | 10.50 ± 0.45 | |
| | | 1.95 ± 0.08 | 9.95 ± 0.43 | 10.50 ± 0.25 |
| | 2-methyl-2-butanol | 2.39 ± 0.09 | 8.61 ± 1.50 | |
| | | 2.92 ± 0.09 | 10.50 ± 1.78 | |
| | | 2.25 ± 0.09 | 8.12 ± 1.34 | |
| 4MCHexOH + OH | Propene | 0.64 ± 0.01 | 17.10 ± 2.59 | |
| | | 0.76 ± 0.03 | 20.30 ± 3.19 | |
| | | 0.76 ± 0.02 | 20.40 ± 3.10 | 18.70 ± 1.42 |
| | Cyclohexene | 0.27 ± 0.01 | 18.20 ± 4.55 | |
| | | 0.27 ± 0.01 | 18.40 ± 4.62 | |
| | | 0.27 ± 0.01 | 18.00 ± 4.46 | |

| Reaction | Reference | $(k_{MSA}/k_R) \pm 2\sigma$ | $(k_{NO3} \pm 2\sigma)/10^{-15}$ | $\bar{k}_{NO3} \pm 2\sigma /10^{-15}$ |
|---|---|---|---|---|
| 4MCHexOH + $NO_3$ | 1-butanol | 1.08 ± 0.12 | 3.39 ± 1.11 | |
| | | 1.81 ± 0.15 | 5.70 ± 1.82 | |
| | | 0.79 ± 0.07 | 2.51 ± 0.80 | 2.69 ± 0.37 |
| | 2-ethyl-1-hexanol | 0.71 ± 0.10 | 2.08 ± 0.72 | |
| | | 1.00 ± 0.10 | 2.93 ± 0.96 | |
| | | 0.86 ± 0.08 | 2.52 ± 0.82 | |

[a]The uncertainties for rate coefficients of MSA ($\sigma_{KMSA}$) were calculated from the uncertainty of slope of plots ($\sigma_{slope}$) and the uncertainty of the reference ($\sigma_{KR}$) by using the propagation of uncertainties. [b]Weighted average according to the equation $(w_1 k_1 + w_2 k_{2+} \ldots)/(w_1 + w_2 \ldots)$; ($w_i = 1/\sigma_t^2$). The uncertainty of weighted average ($\sigma$) was given by $(1/w_1 + 1/w_2 + \ldots)^{-0.5}$

RC: (xi) Table 6: Total C balance for some reactions (particularly for Cl reactions) is extremely low. Please explain.

AC: Table 6 has been revised and it has been observed that there were some erroneous data in the calculation of the carbon balance, although it is practically similar to the initial ones.
The very low values of the total balance of C could be explained because, as it is observed in the residual spectra, after eliminating all bands of the known compounds, there are still absorption bands of compounds that couldn't be identify or quantify, since they are not commercial. These compounds could correspond to the hydroxycarbonyls and dialcohols compounds shown in the different reaction schemes.

**Technical issues:**
**The language in some parts of the manuscript could be improved. I've noticed some typos and other technical issues throughout the manuscript which are listed below. I have not thoroughly checked for technical issues in supplementary material and I'd request the authors to review this section again.**
RC:(i) Title: Some words of the title are written in title case while other words are not. Consistency should be maintained.
AC: This has been corrected.
RC:(ii) Abstract: P1 L13 – is the full stop at the end of this line valid? It looks like the sentence is continuing in the next line.
AC: By suggestion of referee#2 this part of abstract has been modified. In the new version of manuscript, the next sentences appear:

"Rate coefficients (in $cm^3$ molecule$^{-1}$ s$^{-1}$ unit) measured at $\sim$ 298K and atmospheric pressure (720 $\pm$ 20 Torr) were as follows: $k_1$ (E-4-methyl-cyclohexanol + Cl) = (3.70 $\pm$ 0.16) $\times$ 10$^{-10}$, $k_2$ (E-4-methyl-cyclohexanol + OH) = (1.87 $\pm$ 0.14) $\times$ 10$^{-11}$, $k_3$ (E-4-methyl-cyclohexanol + NO$_3$) = (2.69 $\pm$ 0.37) $\times$ 10$^{-15}$, $k_4$ (3,3-dimethyl-1-butanol + Cl) = (2.69 $\pm$ 0.16) $\times$ 10$^{-10}$, $k_5$ (3,3-dimethyl-1-butanol + OH) = (5.33 $\pm$ 0.16) $\times$ 10$^{-12}$, $k_6$(3,3-dimethyl-2-butanol + Cl) = (1.21 $\pm$ 0.07) $\times$ 10$^{-10}$ and $k_7$(3,3-dimethyl-2-butanol + OH) = (10.50 $\pm$ 0.25) $\times$ 10$^{-12}$."

RC: P1, L20: when you first introduce HCOH, write its full chemical name.
AC: This has been corrected.
RC: (iii) P2, L25: Change "Others" to "Other".
AC: This has been corrected.
(iv) P3, L23: the rate coefficient for the reaction with MSA is termed as kS here while in equation (1) it is termed as kMSA. Please, correct. Also, define kS and kR here.
AC: This has been corrected.
(v)  P4, L5: equation (1) is written in Roman (I). Please change.
AC: This has been corrected.
(vi) P4, L5: 1st and 2nd brackets are wrongly placed for both the terms.
AC: This has been corrected.
(vii) In many places, hydrogen "subtraction" is written instead of "abstraction".
AC: The text has been revised and we have decided to use the term "subtract" to the analysis procedure of FTIR spectra and "abstract" to indicate the elimination of an hydrogen of MSA by the oxidants.
RC: (viii) P6, L30: add "for MSA" after ": : : NO3 are higher".
AC: This has been corrected.
RC: (ix) P7, L2: write the full word "molecule", not the abbreviation "molec" in the unit.
AC: This has been corrected.
RC: (x) P7, L10: change "not" to "no".

AC: This has been corrected.

RC:(xi) P8, L1: "develop" can be changed to "developed".

AC: This has been corrected.

RC: (xii) P8, L19-20: check Units.

AC: The units have been checked

RC: (xiii) P8, L23-24: These values could be included in Table 2.

AC: These values and the ratio of $k_{exp}/k_{log}$ have been included in Table 2 in the new version of the manuscript.

RC: (xiv) P8, L36: "Bands that are due: :" – sentence is incomplete.

AC: This has been corrected.

RC: (xv) P9, L15: "of Fig. 3" can be changed to "in Fig. 3".

AC: This has been changed.

RC: (xvi) P11, L4 (and in other places): "stablish" can be replaced by "establish".

AC: All words have been replaced.

RC: (xvii) P14, L22: change "MSA have not a" to "MSA do not have a".

AC: This has been changed.

RC: (xviii) Table 2: "Ratio" – "R" is capital in one place and small in the other two places.

AC: This has been changed.

RC: (xviv) Fig.1 B): Left axis – correct problem with 1st and 3rd brackets.

AC: This has been corrected.

RC: (xvv) Fig2. Title: change "4-methylcyclohexanona" to "4-methylcyclohexanone".

AC: This has been corrected.

RC: (xvvi) Fig3. Picture quality is poor. Axis fonts are not readable.

AC: The Fig.3 has been replaced.

RC: (xvvii) Fig.4: the unit of x-axis missing. Describe the terms used in both the axis.

AC: This has been corrected.

RC: (xviii) Fig7 & 8: the dot sign of radical is missing in some places.

AC: The figures 5, 7 and 8 have been checked. All dot signs of the radicals have been included and their size have been increased.

**Response to the anonymous referee #2**

In the following, the referee's comments (RC) are reproduced (black) along with our replies AC (blue) and changes made to the text (red) in the revised manuscript.

**Title: Atmospheric fate of a series of Methyl Saturated Alcohols (MSA): Kinetic and Mechanistic study**

**General comments:** This paper reports the experimental studies on the reactions of Cl atoms, OH radicals and NO3 with MSA using Relative rate method using FTIR and GC-TOFMS as analytical tools. They have carried out the product analysis at room temperature in presence of synthetic air and reported the products obtained for the title reactions.

**RC**

**Recommendation: This work is good and carried out systematically but, of routine nature not suitable to ACP and can be published in more specific journals related to kinetics. However, the authors may consider the following suggestion to improve the quality of the Paper, if they wish to submit to another specific journal.**

AC:

The authors, before deciding to send the work to the ACP, have evaluated if the work could be framed within any of the themes of the journal finding that the work is effectively framed in the thematic areas and activities presented in this journal:

-Subject area: gases. Effectively, our work focuses on gas phase reactions

-Research activity: Laboratory study. Our study is an experimental work carried out in the laboratory.

-Altitude range. The studied reactions have an interest at the troposphere level since the compounds under study are emitted to the troposphere by different processes, and taking into account the possible future use of these compounds as additives to the fuels, this use could cause significant troposphere emissions.

-Sciences focus. Our work corresponds clearly to the field of the Chemistry.

Therefore, we consider that our work is perfectly publishable in the ACP. It presents the first study of reaction products with proposals of reaction mechanisms that provides a valuable information on the reactivity in the troposphere of compounds of atmospheric interest such as are the alcohols. Alcohols are being object of study by the scientific community and our work provides the first data about the atmospheric reactivity of a series of alcohols that in a future could be used as fuel. Furthermore, our work helps to the scientific community in particular and to the society in general to understand the behavior and implications of alcohols in the atmosphere. The authors know that there are other journals where this work can be published but we consider that ACP is the best for two reasons mainly:

1-This journal allows more extensive works than others, since it does not limit the number of pages. In our case, this fact is important since the work involves discussing a lot of experimental results.

2-It is a journal of wide diffusion in the environmental field with open access and not exclusive of kinetics. The publication of our work in a specific journal of kinetic would imply that diffusion of our results was more restricted.

**Major issues regarding the manuscripts:**

**1-RC:** The manuscript is difficult to read and understand, confusing in many places, careful reading should be done throughout.

   **AC:** The manuscript has been reread. Some sentences or paragraphs in the kinetic discussion section have been rewritten in order to better understand of work presented. All modifications have been indicated in the new version of manuscript.

**2-RC:** In the abstract, "$(2.70 \pm 0.55) \times 10^{-10}$ and $(5.57 \pm 0.66) \times 10^{-12}$ for reaction of 3,3-dimethyl- 1-butanol with Cl and OH radical respectively and $(1.21 \pm 0.37) \times 10^{-10}$ and $(10.51 \pm 0.81) \times 10^{-12}$ for reaction of 3,3-dimethyl-2-butanol with Cl and OH radical respectively". – sentence should be rewritten.

**AC**: This sentence been rewritten. We have added the following text:

"Rate coefficients (in $cm^3$ molecule$^{-1}$ s$^{-1}$ unit) measured at ~ 298 K and atmospheric pressure $(720 \pm 20$ Torr) were as follows: $k_1$ (E-4-methyl-cyclohexanol + Cl) = $(3.70 \pm 0.16) \times 10^{-10}$, $k_2$ (E-4-methyl-cyclohexanol + OH) = $(1.87 \pm 0.14) \times 10^{-11}$, $k_3$ (E-4-methyl-cyclohexanol + NO$_3$) = $(2.69 \pm 0.37) \times 10^{-15}$, $k_4$ (3,3-dimethyl-1-butanol + Cl) = $(2.69 \pm 0.16) \times 10^{-10}$, $k_5$ (3,3-dimethyl-1-butanol + OH) = $(5.33 \pm 0.16) \times 10^{-12}$, $k_6$ (3,3-dimethyl-2-butanol + Cl) = $(1.21 \pm 0.07) \times 10^{-10}$ and $k_7$ (3,3-dimethyl-2-butanol + OH) = $(10.50 \pm 0.25) \times 10^{-12}$ "

**3-RC**: In page nº.2; "Therefore, previously to the massive use, it is necessary to study the reactivity of the large alcohols in atmospheric conditions, in order to establish and to evaluate their atmospheric impact". – the atmospheric conditions may vary depends upon the altitude hence temperature dependent and pressure dependent studies need to be done in order to get the complete atmospheric impact.

**AC:** We agree with the referee that reactions must be evaluated in the temperature range typical of the Troposphere, but it is not possible in our case because our experimental system does not allow us to work at different temperatures. We are working in a modification of the reactors to do these experiments in a near future. In the case of the effect of pressure on rate coefficients, the literature data about kinetic studies of this kind of reactions, show not significant influence on the rate coefficients, in the typical range of atmospheric pressure. Kinetic studies about Oxygenated Volatile Organic Compounds in which it is used absolute method (that works in different ranges of pressures) and relative method (that works at atmospheric pressure) the rate coefficients obtained are similar taking into account the experimental errors. For this reason, the most of the reactions of atmospheric interest are evaluated only at atmospheric pressure (https://kinetics.nist.gov/kinetics/index.jsp).

The authors with this phrase, do not mean that with our results the atmospheric implications of alcohols are perfectly established, but help to establish these implications. The authors know that more experiments should be carried out to obtain a complete knowledge of the atmospheric implications of these alcohols, as indicated in the conclusion section. A phrase indicating the need to perform the kinetic study at typical atmospheric temperatures in order to obtain more information of the reaction mechanism and can extrapolate the data of rate coefficients to other typical atmospheric conditions, could be added in the conclusion section.

"...However, kinetic experiments in the tropospheric temperature range are necessary to obtain more information about the reaction mechanism and extrapolate the data of rate coefficients to other typical atmospheric conditions and thus be able to better establish the atmospheric impact of the alcohols."

**4-RC:** What are the limits for photolysis and wall effect limits? Is there any preliminary reaction carried out to check secondary chemistry for the title reactions? Explain.

**AC**: As it is indicated in the main text (kinetic study section), previous to the kinetic study, it is habitual to carry out a series of experiments in order to establish the possible secondary reactions. In the case of MSA, the next experiments were done.

-Checking dark reactions of MSA together with the reference compounds.

-Checking the reaction of MSA and refence compounds with $Cl_2$, $CH_3ONO$, NO and $NO_2$.

-Checking the photolysis of MSA and reference compounds.

In all cases, these experiments showed insignificant loss processes of reactants.

**5 RC**: Page 4, "Kinetic measurements were performed at room temperature 298 K) and atmospheric pressure ~ (720 Torr)" – Authors have stated the pressure at which the reactions were carried out is 720 Torr throughout the main text but, in the abstract it is stated as 740 Torr. It is advised to check the values.

**AC:** The experiments on Teflon® reactor were done at atmospheric pressure that each day could variated between 700 and 710 Torr. The experiments on Pyrex® reactor were done at pressure between 710 and 740 Torr. In the new version of the manuscript, all data of pressure have been changed by $720 \pm 20$ Torr.

**6 RC**: CH3ONO was synthesized in the laboratory – give the procedure and specify the purity of the prepared compound with NMR, IR etc.

**AC:** The procedure of synthesis of $CH_3ONO$ is the same that the described by Taylor et al 1980 and we consider that it is not necessary to give details of the synthesis procedure in order to not extent more the manuscript. When the $CH_3ONO$ is synthetized, an IR spectrum is done and it is compared with the $CH_3ONO$ reference spectrum of database. In all cases a high purity is observed.

An example of IR spectrum of $CH_3ONO$ synthetized and reference spectrum from Eurochamp 2020 database (https://data.eurochamp.org/data-access/ir-spectra/) is showed in the figure above.

[Figure]

**7-RC:** Page 5, "During the reaction process in the 50 L Pyrex® glass chamber, the identification of products was made using the FTIR analysis but, at the same time, a sample was taken and analyzed in the SPME/GC-TOFMS system". - is quite confusing and should be rephrased for better understanding for the readers.

**AC:** This sentence will be rewritten.

"In some experiments carried out in the 50 L Pyrex® reactor, a simultaneous identification of products was performed using both detection techniques. For that, one sample of mixing reaction was taken from this reactor using the SPME and subsequent analyzed with GC-TOFMS"

**8 RC**: Page 5, "To obtain the yield percentage of carbon, the yield obtained is multiplied by 100 and by the ratio of carbons between the product and the MSA from which it comes".
– not clear.

**AC**: This sentence will be removed in the new version of manuscript and a footnote in the table 6 will be included indicating how the total carbon has been calculated.

$$Total\ Carbon\ (\%) = \sum_{1}^{i} \left( \frac{n^{\circ}\ of\ carbon\ of\ product_i}{n^{\circ}\ of\ carbon\ of\ MSA} \times molar\ yield_i\ (\%) \right)$$

**9-RC**: Authors are advised to use the recommended rate coefficients for all the reference reactions for better reliability of the rate coefficients.

**AC:** The authors know that is convenient to use the recommended rate coefficients for all the reference compounds, but in some cases it was not possible to use reference compounds with a recommended rate coefficient, because their IR bands overlapped with the IR bands of MSA or because there was no other reference compound with rate coefficients similar to the MSA. So we had to use reference compounds whose rate coefficients were well established but not as well as recommended compounds. In order to assure the rate coefficients determined, different reference compounds were used.

**10-RC:** Page 7, "This behavior could be explained for the different size and electronic properties   of each oxidant that make the Cl atom the most reactive (value of k in the limit of collision)  but also less selective than OH and NO3 radicals". – needs more explanation.

AC: More explanation has been included in the manuscript.

"This behavior could be explained considering the geometry and the electronic density of each oxidant, together with the kinetic Collision Theory. As Cl atom has spherical distribution of its density, for the collision any orientation is adequate, in addition the Cl atoms presents less steric hindrance. Then, comparatively the Cl reaction is less selective and faster with values for the rate coefficients, k, in the collision limit. However, the OH radical presents an asymmetric electron density located mostly over its oxygen atom. Therefore, for the OH reaction the oxygen of OH radical, must be specific oriented to the hydrogen of the MSA that will be abstracted. The electronic density of nitrate radical is distributed around the three oxygens which implies several appropriate orientations. However, as the nitrate radical has a non-linear structure, the steric hindrance is much bigger than for the OH and it reduces the reactivity of $NO_3$ in relation to those of OH radical."

**11-RC:** Page 7, "In the case of 3,3-dimethylbutanols, there is………… of the structure of the organic compound on the reactivity (SAR Method, Kwok and Atkinson, 1995)". -Rewrite the sentence.

**AC:** Taking in account your suggestion and the suggestion of referee #1, this part of the kinetic discussion has been rewritten in order to more clarity. In the new version of manuscript, the next sentences appear:

"The activating effect of hydroxyl group of the alcohols was quantified by different authors (Kwok and Atkinson 1995; Kerdouci et al, 2010, Calvert et al. 2011) taking into account the available kinetic data reported in bibliography, obtaining the factor of reactivity for the hydroxyl group, F(-OH)). This factor of reactivity is different for each oxidant, 1.18 for Cl reaction, 2.35 for reaction with OH (Calvert et al. 2011) and 18 for $NO_3$ reaction (Kerdouci et al., 2010). There are no data of rate coefficients for the reactions of the homologous alkanes of the MSA studied in this work with $NO_3$ radical, and therefore it is not possible to check out the effect of hydroxyl group in the reactivity of $NO_3$ reaction. However, according to the factor of reactivity obtained by Kerdouci et al. (2010) for the reactions of alcohols with $NO_3$, this effect is higher than the corresponding to Cl and OH reactions."

**12-RC**: Page 7, "The activating effect of the length chain in the reactivity is being more marked in the Cl reaction than in the case of OH and NO3 reactions". Why? Proper explanation should be given. Sentence is very confusing.

**AC:** To clarify, this sentence will be rewritten as follows:

"The activating effect of the length chain in the reactivity of alcohols is more evident in Cl reactions than OH reactions (See Table S1). Furthermore, if the rate coefficients of 3-methyl-1-butanol (3M1ButOH) and 3,3DM1ButOH with Cl and OH reactions are compared, it can be observed a slight increase of rate coefficient for Cl reaction ($k_{3M1BuOH+Cl}$ = 25.0 × $10^{-11}$; $k_{3,3DM1ButOH+Cl}$ = 26.9 × $10^{-11}$) and an important decrease of the rate coefficient for OH reactions ($k_{3M1BuOH+OH}$ = 14 × $10^{-12}$; $k_{3,3DM1ButOH+OH}$ = 5.33 × $10^{-12}$). This behavior could be explained by the different order of reactivity between the oxidants. For Cl atom, more reactive (k order of 10⁻

$^{10}$ cm$^3$ molecule$^{-1}$ s$^{-1}$) but less selective, an increase of the length chain or in the number of methyl groups implies more hydrogens available to be abstracted and therefore an increase of the rate coefficient. However, for OH radicals, less reactive and more selective, the attack for H-abstraction will be carried out in a specific place, so an increase of the chain has not a significative effect to the reactivity, even the presence of a second methyl group disfavor the reaction probably due to the steric hindrance near to the attack position."

**13-RC**: Page 8, "In general, the SAR method applied to alcohols predicts better rate coefficients for Cl atoms and OH radical than for NO3 radical, especially for primary alcohols". – But the value of the kexp/kSAR for the reaction of 3,3DM1ButOH with NO3 is found out to be 3.29. Please give the explanation for this discrepancy.

**AC:** As it is indicated in the section of reaction product study, the reason of the discrepancy could be the fact that the SAR method developed for NO$_3$ reaction by Kerdouci et al. does not consider the effect of the factor -CH$_2$-OH in the reactivity. Moreover, the authors of this SAR method (Kerdouci et al. 2010, 2014) indicate the smaller predictive ability of this SAR method for saturated alcohols with NO$_3$ due to the lack of experimental data.

**14-RC**: Page 8, "….and in some cases due to heterogeneous reactions with the walls of the gas cell". – contradicting statement - check the experimental method given!

AC: In this case the heterogeneous reactions, it refers to reactions of the precursors with the Pyrex walls of gas cell. In the text an annotation will be included. It is the following:

"..and in some cases are due to heterogeneous reactions of these precursors with the walls of the gas cell'

**15-RC:** Page 10, why no exploration on OH + NO and NO2 + NO3??

AC: We don't understand exactly what you mean with this question. Our experimental procedure does not allow us to carry out experiments in absence of NO in the case of OH reactions and in absence of NO$_2$ in the case of NO$_3$ reaction.

**16-RC:** "The kinetic and product study confirms that the atmospheric degradation mechanism 1 for methyl saturated alcohols and possibly for the rest of unstudied saturated alcohols, proceeds mainly by abstraction of the hydrogen atom bonded to carbon instead hydrogen atoms bonded to oxygen atom of the alcohol group". – This is a known fact and should be removed from the conclusion.

AC: This assumption has been included in the document because we would like to remark that effectively our results support, confirm, the general reactivity of alcohols.

In the new version of manuscript, the first and second conclusions have been reorganized as follows.

"The kinetic and product study support that: 1 -The atmospheric degradation mechanism for MSA, and possibly for the rest of unstudied saturated alcohols, proceeds by abstraction of the hydrogen atom bonded to a carbon instead of hydrogen atoms bonded to the oxygen atom of the alcohol group. 2 -The reaction mechanism in the H-abstraction process depends on the oxidant.

Chlorine atoms abstract any type of alkyl hydrogen from saturated alcohols with a high percentage, compared to the hydroxyl radical and the nitrate radical. OH and NO$_3$ radicals abstract mainly the hydrogen in the $\alpha$ position, if the saturated alcohols are secondary. For primary alcohols, the abstraction of a hydrogen in $\beta$ position could be also important in the reaction with NO$_3$ radical. Therefore, more kinetic studies for NO$_3$ radical with primary alcohols are necessary to update the SAR method developed by Kerdouci et al., and to quantify the effect of the OH group in $\beta$ position, (-CH$_2$OH)."

However, if the reviewers consider this sentence unnecessary could be eliminated of the conclusions.

**17-RC**: Main text and Table 2 values – The given reasons are different. Please clarify.
AC: The main text and table 2 have been checked. This table has been modified as the referee #1 suggests. In the new version of manuscript this section has been modified.

"…The estimated rate coefficients, $k_{log}$, according with Eq (2) and Eq (3), and the ratios ($k_{exp}/k_{log}$), are also shown in Table 2. This estimation method obtains slightly better rate coefficient for 3,3DM1ButOH + NO$_3$ reaction ($k_{exp}/k_{log}.$ = 1.53) than SAR ($k_{exp}/k_{SAR}$ = 3.24). However, for Cl reactions the ratios $k_{exp}/k_{log}$ are in the range of 0.6-1.97, indicating that the Eq (2) predicts worse the rate coefficients than SAR method. Again, this fact could be due to the different mechanism reaction in the H-abstraction process for Cl and OH reactions. Such as it has been indicated above to apply these relationships both oxidants must react according to the same mechanism...."

**18-RC**: Since 2-butanol is not a suitable reference authors could have been chosen another reference for their studies.
AC: We suppose that you want to say 1-butanol.
As it has been explained above in question 9, it is difficult to find reference compounds with the necessary characteristic to be used in these experiments.

**19-RC:** In Table 1, it seems like authors have taken the average of deviation values obtained in individual rate coefficients (column 4). It is advised to carry out the proper analysis of the errors by standard error propagation method. (For reference see *Chem. Phys. Lett*. 2013, 590, 221-226 and *New J. Chem*. 2017, 41, 7491-7505).
AC: By suggestion of referee #1 the data of Table 1 have been revised. The analysis of data have been done as it is explained in the main text and footnote of Table 1 of the new manuscript version.

"The ratios of the rate coefficients, $k_{MSA}/k_R$, the absolute rate coefficients and the weighted average are shown in Table 1. The error of $k_{MSA}/k_R$ are given by 2 times the statistical deviation calculated from the least-square fit of the plot of Eq. (1). The uncertainties for rate coefficients of MSA ($\sigma_{kMSA}$) were calculated from the uncertainty of slope of plots ($\sigma_{slope}$) and the uncertainty of the reference ($\sigma_{kR}$) by using the propagation of uncertainties. The average value of the rate coefficient obtained with different reference compounds and its associated error were obtained by weighted average."

**20-RC**: Why the effect of the bath gas on the rate coefficients were not explored?

AC: The effect of the bath gas on the rate coefficients were explored in a previous study (Cabañas et al. 2005). The results of the rate coefficients in $N_2$ and air where similar, considering the experimental error. So, we always use $N_2$ as bath gas (except to the OH reaction because it is necessary $O_2$ to generate the OH) because it is less expensive than air.

Relevant changes.

The pages and lines correspond with the modified manuscript.

-Pag. 1. Lines 10. The keywords have been included
-Pag 1. Lines 13-17. This paragraph has been included.
-Pag 3. Lines 19-20. The figure has been modified.
-Pag. 4 lines 11-37. The description of the experimental systems has been modified including more details respects to the initial version of manuscript.
-Pag. 5 Lines 10-13. The sentence has been rewritten.
-Pag. 6 lines 25-31. More information about the uncertainty's treatment has been included.
-Pag. 7 lines 11 to pag 9 line 24. The discussion of kinetic data has been rewritten to a better understanding.
-Pag. 7 lines 30-32. These lines have been eliminated and an equation for calculation of Carbon balance has been included as a footnote in Table 6.
-Pag. 11. Lines 16-20. This paragraph has been included to explain better the different yields of E-4-methylcyclohexanone obtained for both experimental systems.
-Pag. 15. Lines 24-32. Two conclusions have been reorganized in one.
-Pag 16. Lines 24-27. A new paragraph has been included.
-Pag.20 Lines 1-3. A reference has been included.
-Pag. 21. Lines 19-21. A reference has been included.
-Pag. 23. Table 1. Has been modified including the data of relative rate coefficients, eliminating the column 4 of the initial Table 1 and recalculating the errors of average value of the rate coefficient using the weighted average
-Pag. 24 Table 2. Has been modified including more data.
-Pag. 30 to Pag 37. The figures 1, 2,3,4,5,6,7,8, have been modified slightly with a better resolution in some case or to include some missed element in other.

[revised manuscript text omitted]

3,3DM1ButOH=1.22 × 10⁻¹⁵; k_NO3-3,3DM2ButOH=2.48 × 10⁻¹⁵ and k_NO3-4MCHexOH=4.81 × 10⁻¹⁵. This estimation method obtains slightly better rate coefficient for 3,3DM1ButOH + $NO_3$ reaction ($k_{exp}/k_{log}$. = 1.53) than SAR ($k_{exp}/k_{SAR}$ =

3.24). However, for Cl reactions the ratios $k_{exp}/k_{log}$ are in the range of 0.6-1.97, indicating that the Eq (2) predicts worse the rate coefficients than SAR method. results than SAR for NO₃ reactions. The better prediction for the

NO₃ rate coefficients than for those of Cl could be due to the Again, this fact could be due to that the mechanism for Cl atom reactions is the different mechanism reaction in the H-abstraction process for Cl and than for OH

radical reactions. Such as it has been indicated aboveAssumption that must be satisfied 
[revised manuscript text omitted]

**S2. Structure-Activity Relationship (SAR) method**

SAR method allows to estimate a rate coefficient of an organic compound from its structure. The only possibility of the reaction of the studied compounds in this work with the atmospheric oxidants is the abstraction of an hydrogen atom. Consequently, the estimated rate coefficients of MSA are obtained from the sum of the rate coefficients for the H-atom abstraction from the primary ($k_{prim}$ (CH$_3$- )), secondary ($k_{sec,}$ (-CH$_2$-)) and tertiary ($k_{tert,}$ (-CH<)) groups and from the alcohol ($k_{OH}$ (-OH)) group, taking into account the influence of the substituents attached to these groups, through substituent factors F(X), F(Y) and F(Z) (Equation S4).

$$k_{abs} = \sum k_{prim}F(X) + \sum k_{sec}F(X)F(Y) + \sum k_{tert}F(X)F(Y)F(Z) + \sum k_{OH} \quad (S4)$$

At 298K rate coefficients for H-atom abstraction (in units of cm$^3$molecule$^{-1}$s$^{-1}$) and the reactivity factor for the reaction with OH are $k_{prim}$ = 1.36 × 10$^{-13}$; $k_{sec}$ = 9.34 × 10$^{-13}$; $k_{tert}$ = 1.94 × 10$^{-12}$ and $k_{OH}$ = 1.4 × 10$^{-13}$; F(CH$_3$) = 1; F(-CH$_2$-) = F(-CH<) = F(>C<) = 1.23 and F(-OH) = 3.5 from AOPWIN. The parameters for the reaction with Cl atoms are $k_{prim}$ = 2.84 × 10$^{-11}$; $k_{sec}$ = 8.95 × 10$^{-11}$; $k_{tert}$ = 6.48 × 10$^{-11}$ (in units of cm$^3$molecule$^{-1}$s$^{-1}$); F(CH$_3$) = 1; F(-CH$_2$-) = F(-CH<) = 0.8 and F(-OH) = 1.18 from Calvert et al. 2011. By last, the parameters used for the reaction with NO$_3$ radicals are $k_{prim}$ = 1 × 10$^{-18}$; $k_{sec}$ = 2.56 × 10$^{-17}$; $k_{tert}$ = 1.05 × 10$^{-16}$ and $k_{OH}$ = 2 × 10$^{-17}$ (in units of cm$^3$molecule$^{-1}$s$^{-1}$); F(CH$_3$) = 1; F(-CH$_2$-) = 1.02; F(-CH<) = 1.61; F(>C<) = 2.03 and F(-OH)=18 from Kerdouci et al. 2010, 2014.

The calculations for 4-methyl-cyclohexanol are the following:

$k_1$ = $k_{tert}$×F(-OH)×F(-CH$_2$-)×F(-CH$_2$-)

$k_2$ = $k_3$ = $k_5$ = $k_6$ = $k_{sec}$×F(-CH<)×F(-CH$_2$-).

$k_4$ = $k_{tert}$×F(-CH$_3$)×F(-CH$_2$-)×F(-CH$_2$-)

$k_7$ = $k_{prim}$×F(-CH<)

$k_{4MCHexOH+Cl}$ = 3.42×10$^{-10}$ cm$^3$molecule$^{-1}$s$^{-1}$

$k_{4MCHexOH+OH}$ = 1.92×10$^{-11}$ cm$^3$molecule$^{-1}$s$^{-1}$

$k_{4MCHexOH+NO3}$ = 2.27×10$^{-15}$ cm$^3$molecule$^{-1}$s$^{-1}$

The calculations for 3,3-dimethyl-1-butanol are the following:

$k_1$ = $k_{sec}$×F(-OH)×F(-CH$_2$-)

$k_2$ = $k_{sec}$×F(>C<)×F(-CH$_2$-)

$k_3$ = $k_4$ = $k_5$ = $k_{prim}$×F(>C<)

$k_{3,3DM1ButOH+Cl}$ = 2.10×10$^{-10}$ cm$^3$molecule$^{-1}$s$^{-1}$

$k_{3,3DM1ButOH\ OH}$ = 6.08×10$^{-12}$ cm$^3$molecule$^{-1}$s$^{-1}$

$k_{3,3DM1ButOH\ +NO3}$ = 0.55×10$^{-15}$ cm$^3$molecule$^{-1}$s$^{-1}$

The calculations for 3,3-dimethyl-2-butanol are the following:

$k_1 = k_{prim} \times F(-CH<)$

$k_2 = k_{tert} \times F(>C<) \times F(-CH_3) \times F(-OH)$

$k_3 = k_4 = k_5 = k_{prim} \times F(>C<)$

$k_{3,3DM2ButOH+Cl} = 1.52 \times 10^{-10}$ cm$^3$molecule$^{-1}$s$^{-1}$

$k_{3,3DM2ButOH\ OH} = 9.16 \times 10^{-12}$ cm$^3$molecule$^{-1}$s$^{-1}$

$k_{3,3DM2ButOH\ +NO3} = 3.86 \times 10^{-15}$ cm$^3$molecule$^{-1}$s$^{-1}$

**Tables**

[revised manuscript text omitted]

-**Fig. S76.** EI MS spectra of the peaks of chromatograms shown in Fig. S6 obtained for the reaction of of 3,3DM1ButOH 3,3-dimethyl-1-butanol with Cl, Cl + NO, HO and NO₃. (a) $t_R$= 6.00 min; (b) $t_R$= 8.61 min; (c) $t_R$= 13.17 min; (d) $t_R$= 5.05 min).

**Fig. S87**. Reaction mechanism for degradation of 3,3-dimethylbutanal with the atmospheric oxidants in presence of NOx. H-Atom aAbstraction from the -COH gGroup in 3,3-dDimethylbutanal. Aschamnn et al., 2010.

[Figure]

**Fig. S98**. A) FTIR spectra obtained in the reaction of 3,3DM2ButOH with Cl (a), Cl + NO (b), OH (c) NO$_3$ (d) at 5 minutes of reactions. (e) FTIR reference spectrum of 3,3-dimethyl-2-butanonea. B) FTIR spectra obtained in the reaction of 3,3-dimethyl-2-butanol with Cl (a), Cl + NO (b), 25 minutes and 35 minutes of reactions respectively. (c) IR PAN spectrum. C) Residual FTIR spectra after subtraction of all known bands. Cl (a), Cl + NO (b), HO (c) and NO$_3$ (d).

[Figure]

**Fig. S109.** SPME/GC-TOFMS chromatograms obtained for the reaction of of 3,3DM2ButOH  with Cl, Cl + NO, HO and NO₃, and reference chromatograms of 3,3DM2ButOH and 3,3-dimethyl-2-butanone.

| $t_R$ (min) | EI MS |
|---|---|
| 2.16 |
[Figure]
 Acetone |
| 5.39 ? | 2,2-dimethylpropanal |
| 6.03 | 3,3-dimethyl-2-butanone |
| 6.22 | Acetic Acid (SI 80%) |
| 6.46 | SPME |
| 6.96 | Nitrated compound |
| 7.75 | |
| 8.37 | |

[Figure]

Fig. S11<s>0</s>. EI MS spectra of the peaks of chromatograms shown in Fig. S10 obtained for the reaction <s>of</s> of 3,3DM2ButOH with Cl, Cl + NO, HO and NO₃.

A)

[Figure]

(B)

[Figure]

**Fig., S12₁.** Concentration-time profiles of 3,3DM2ButOH and reaction products formedobtained for the reaction of 3,3DM2ButOH with Cl atoms in absence (A) and presence of NO (B).

[Figure]

**Fig. S132.** Reaction mechanism for degradation of 3,3-dimethyl-2-butanone with the atmospheric oxidants in presence of NOx.

---

## Author Response (AR2)

In the following, editor comments (EC) are reproduced (black) along with our replies AC (blue) and changes made to the text (red) in the revised manuscript.

Editor comments:

However, the presentation of the work falls below normal publication standards. The overall quality and clarity of the presentation greatly hinders the communication of the scientific results.

On the basis of the reviewer comments and my review, I am requesting that the authors make major revisions to their manuscript to improve the communication, clarity, logic, (remove) unnecessary repetition ("As mentioned above" type text), (remove) meaningless general statements, English, and grammar. In addition, the presentation of the scientific results for the molecules under study could be made more concise and earlier to follow for the reader. The tables and figures seem fine.

AC: We thank the editor for the interest shown on our work and the comments and suggestions.

The manuscript has been revised and unnecessary repetitions have been removed. We expect that with the english revision the meaningless statements have been eliminated.

The authors have tried to make a presentation of the scientific results more concise but we consider that this is a work with many experiments and results, and it is difficult for us to reduce the work without loss of scientific rigor. We know that in the kinetic discussion section there are information relative to general topics of reactivity of alcohols that could be omitted (page 6 lines 28-38 and page 7, lines 1-3 of new manuscript.pdf) because this information is well established and known. The authors consider that given the scope and dissemination of the ACP, more details would help to anyone who was not an expert in the subject to understand better the work. Indeed, at the suggestions of both referees, more information and explications had to be included. However, if you consider that this information is not necessary to understand better the discussion of the kinetic results, we can eliminate it.

The authors would appreciate if you could tell us which parts should be reduced or otherwise should be discussed further. In our opinion we consider that this version is fine, although it is possible that there are still some grammatical errors, that would be eliminated in the final review before publication.

I highly recommend that the authors have a native English-speaking colleague critically (line-by-line) review the manuscript prior to re-submission. This is simply too large a task for a reviewer or editor.

AC: The text of manuscript has been sent to " Proof-Reading-Services.com" for american english revision in order to make a rigorous of /vocabulary/grammar/scientific expressions. In order to do not pay a lot of for the revision, abstract, references, tables, figures and Supplementary material have not been sent to english revision.

**EC: A few general theme comments:**

**\*Only one of the compounds included in this study (3,3DM2ButOH) is actually methyl saturated. Therefore, the title and text are in error. This correction would lead to the removal of the misleading MSA acronym.**

AC: All studied alcohols are saturated compounds (they do not have double or triple bonds) and all of them have at least one methyl group, so we considered methyl saturated alcohols to be a good acronym for the studied compounds, that is usual for compounds without multiple bonds. To avoid confusion, the authors have decided to use saturated alcohols generically (SAs)

**EC: * The abstract mentions "tentative estimation of molecular yields" and quotes very large ranges of yields, which are not meaningful. The authors need to be more specific regarding the actual yield values and for what initiation reaction they are obtained from. I believe this information is actually available within the manuscript tables, although hard to follow in the text. Or, is this a problem with not having good standards?**

AC: The individual molar yield of reaction product obtained for each reaction had not been included in the abstract in order to avoid making it very extensive. The phrase "A tentative estimation of molecular yields has been done obtaining the following ranges (25-60) % for 4-methylcyclohexanone, (40-60) % for 3,3-dimethylbutanal and (40-80) % for 3,3-dimethyl-2-butanone." has been removed to avoid confusion, and some new sentences have been included.

"The main products detected in the reaction of SAs with Cl atoms (absence/presence of NOx), OH and $NO_3$ radicals were: E-4-methylcyclohexanone for the reactions of E-4-methyl-cyclohexanol, 3,3-dimethylbutanal for the reactions of 3,3-dimethyl-1-butanol and 3,3-dimethyl-2-butanone for the reactions of 3,3-dimethyl-2-butanol"

"In addition, the molar yields of the reaction products were estimated".

The molar yields of the products are shown in Tables 3-6. In some cases, as 2,2-dimethylpropanal the molar yields could have large uncertainty because the reference spectra used was an FTIR spectra of a similar compound (2-methylpropanal).

**EC * There is a long discussion of reactivity observations that have been well-established through the development of structure activity relationships (SARs) by Atkinson and co-workers. The small data set from this work is probably not sufficient to revise our thinking of SARs. The present work needs to be placed in the proper perspective.**

AC: In the analysis made of the kinetic results obtained in our work, it is observed that the OH group exerts an activating effect that makes the reactivity of the alcohols greater than that of its alkane homologue, but it is also observed that this activating effect is different according to the oxidant. Indeed, these observations are well established through the development of structure activity relationships (SARs) by Atkinson and co-workers.

The authors do not have the intention of reviewing or changing results of the SARs method. On the contrary, what we want to highlight is that our results are in good agreement with those established by the SARs method in the case of reactions of saturated alcohols with Cl atoms and OH radical and in the case of $NO_3$ radical with secondary alcohols.

In order to avoid more confusion, the paragraph of page. 7, lines 31-37 (clean manuscript date 26/10/2019) has been replaced by.

"..These results show that the activating effect of the OH group of the SA is less important for the Cl than with the OH, behavior that agrees with that established by the Structure Activity Relationship (SARs) methods (Kwok and Atkinson 1995; Calvert et al. 2011).."

On the other hand, in the case of the nitrate radical our study shows that in addition to the activating effect of the OH group there is also an activating effect of the $-CH_2OH$ group and that it is necessary to perform more kinetic studies of reactions of primary alcohols with the nitrate radical, in order to establish this factor, since there are currently very few data available in bibliography. This last is indicated in the manuscript as a conclusion.

"..For primary alcohols, the abstraction of a hydrogen atom in $\beta$–position could also be important in the reaction with $NO_3$ radical. Therefore, more kinetic studies for $NO_3$ radical with primary alcohols are necessary to quantify the effect of the OH group in $\beta$–position, (-CH2OH) and to update the SAR method developed by Kerdouci et al."

**EC:\* The introduction contains a great deal of seemingly unnecessary material and background information. Even with that said, it is not made clear why these particular compounds were chosen for study (biofuels?).**

AC: The introduction has been reorganized in order to show better the relevance of compounds studied.

So, in the first paragraph the necessity of using biodiesel as alternative to conventional diesel is remarked. In this same paragraph, it is also shown that alcohol-diesel blends are a good alternative. Initially the alcohols used were alcohols of short chain (methanol, ethanol) but some problems were found due to their low cetane number, high latent heat of vaporization and high resistance to auto-ignition. In order to avoid these problems, high alcohols as propanol, n-butanol, isobutanol, n-pentanol and therefore the alcohols studied in this work, could be a good alternative as additives in the diesel blends. A new reference has been included to support this last.

" Li, F., Yi, B., Song, L., Fu, W., Liu, T., Hu, H., & Lin, Q. Macroscopic spray characteristics of long-chain alcohol-biodiesel fuels in a constant volume chamber. Proceedings of the Institution of Mechanical Engineers, Part A: JPE, 232(2), 195–207. https://doi.org/10.1177/0957650917721336, 2017."

In the second paragraph of introduction, a revision of sources of alcohols and data about concentrations found in the atmosphere is shown. The use of alcohols of long chain as additives for biodiesel fuel could imply an important source of these alcohols in the atmosphere. So, it is necessary to evaluate their atmospheric reactivity and to establish the atmospheric impact of these compounds.

In the next paragraph, a revision of the atmospheric reactivity of short and long alcohols is made, it does remark the absence of kinetic data or about reaction products of the alcohols studied in this work. The last paragraph explains the study that has been carried out and why it has been done.

We expect that this reorganization and correction of the introduction section, allows a better compression and better justification of the developed research.

**LIST OF THE MAIN CHANGES MADE IN THE MANUSCRIPT.**

Apart of the modification due to the English revision, the next modifications have been made at suggestion of the editor in order to more clarity. Pages and lines indicated are related to the manuscript.pdf with date sent of 26_10_2019.

1-Page 1. Lines 1 and 2. The title has been modified. The acronym MSA has been removed, and "methyl saturated alcohols" has been replaced by "saturated alcohols" in all text of the manuscript, tables and figures.

2-Page 1. Line 12. The next sentence has been included:

"These SAs are alcohols that could be used as fuel additives"

3-Page 1. Lines 17-21. The paragraph has been replaced by the following:

"The main products detected in the reaction of SAs with Cl atoms (absence/presence NOx), OH and $NO_3$ radicals were: E-4-methylcyclohexanone for the reactions of E-4-methyl-cyclohexanol, 3,3-dimethylbutanal for the reactions of 3,3-dimethyl-1-butanol and 3,3-dimethyl-2-butanone for the reactions of 3,3-dimethyl-2-butanol"

4-Page 1. Line 21. The sentence has been modified including "..of Cl atoms and OH radicals with.."

5-Page 1. Line 23. The next sentence has been included "In addition, the molar yields of the reaction products were estimated".

6-Page 1. Line 23. The sentence has been modified as follows

"The products detected, indicate a hydrogen atom abstraction mechanism at different sites on the carbon chain of alcohol…"

7-Page 1. Line 31 "Therefore, the use of saturated alcohols as additives in diesel-blends should be considered with caution"

8-Pages 2 and 3. The introduction has been reorganized.

9-Page 2. Line 17. A new reference has been included.

10-Page 7. Lines 31-37. The text has been modified as follows:

"These results show that the activating effect of the OH group of the SA is less important for the reaction with Cl atoms than with the OH radical, behavior that agrees with that established by the Structure Activity Relationships (SARs) methods (Kwok and Atkinson 1995; Calvert et al. 2011)."

12-Page 9. Lines 21, 22. The sentence has been removed.

13-Page 9. Lines 28-31. The sentence has been rewritten to more clarity as follows:

"Some of these compounds are products from the reactions of the SAs with oxidants. They can also be formed by decomposition of the employed precursors (Cl2, CH3ONO and N2O5) and in some cases, by heterogeneous reactions of these precursors with the Pyrex glass reactor walls."

14-Page 10. Lines 32. Part of the sentence has been modified as follows:

"Figure 5A shows the paths that explain the formation of organic compounds (carbonyl, hydroxycarbonyl, etc).."

15-Page 12. Lines 1 and 2 have been modified to more clarity, as follows:

"..presented in Fig. 6, showing that in the absence of NOx the profiles of acetone and formaldehyde have two trends. It indicates that these compounds are formed as primary and secondary products."

16-Page 12. Lines 13-18. The sentence has modified to more clarity, as follows:

"The higher yield of nitrated compounds in the reaction of 3,3DM1ButOH with nitrate radical could indicate an extra formation of nitrated compounds from secondary reactions..".

17-Page 12. Lines 25-27. The paragraph has rewritten to more clarity, as follows:

"As it can see in Table 4, the estimated molecular yields of 3,3-dimethylbutanal (formed by H atom abstraction at the $\alpha$–position of 3,3DM1BuOH) are very similar to the one predicted by the SARs method for the Cl and OH reactions."

18-Page 13. Lines 15-17. The paragraph has rewritten to more clarity, as follows:

"Plots of concentration versus time for formaldehyde, acetone (Fig. S12A) and nitrated compounds in the reactions of Cl in the presence of NOx (Fig. S12B) show profiles with two trends. This type of profile indicates that formaldehyde and acetone could also be formed by degradation of 3,3-dimethyl-2-butanone (Fig. S13)."

19-Page 13. Lines 17-19. The sentence has been moved to Line 14.

20-Page 13. Line 33. The next sentence has been added to more clarity, as follows:

"It could justify the low estimated molecular yield for 3,3-dimethyl-2-butanone."

21-Page 14. Lines 3-4. The paragraph has been eliminated

22-Page 15. Lines 30-32 The sentence has been rewritten as follows:

"Therefore, more kinetic studies of the $NO_3$ radical reaction with primary alcohols are necessary to quantify the effect of the OH group at the $\beta$–position ($-CH2OH$) and to update the SARs method developed by Kerdouci et al."

23-Page 16. Lines 18-20. The sentence has been changed by:

"Therefore, the use of SAs as additives for diesel blends should be controlled, as poor handling could result in high concentrations of these alcohols in the atmosphere."

24-Page 19. Lines 36-38. A new reference has been included:

"Li, F., Yi, B., Song, L., Fu, W., Liu, T., Hu, H., & Lin, Q. Macroscopic spray characteristics of long-chain alcohol-biodiesel fuels in a constant volume chamber. Proceedings of the Institution of Mechanical Engineers, Part A: JPE, 232(2), 195–207. https://doi.org/10.1177/0957650917721336, 2017."

**Atmospheric fate of a series of Saturated Alcohols : kinetic and mechanistic study**

Inmaculada Colmenar[1,2], Pilar Martin[1,2], Beatriz Cabañas[1,2], Sagrario Salgado[1,2], Araceli Tapia[1,2], Inmaculada Aranda[1,2]

[1]Universidad de Castilla La Mancha, Departamento de Química Física, Facultad de Ciencias y Tecnologías Químicas, Avda. Camilo José Cela S/N, 13071 Ciudad Real, Spain

[2]Universidad de Castilla La Mancha, Instituto de Combustión y Contaminación Atmosférica (ICCA), Camino Moledores S/N, 13071 Ciudad Real, Spain

*Correspondence to*: Pilar Martín (mariapilar.martin@uclm.es)

**Keywords.** saturated alcohols; additives;.re,~~ric reactivity.

**Abstract.** The atmospheric fate of a series of saturated Alcohols (SAs) has been evaluated through the kinetic and reaction product studies with the main atmospheric oxidants. These SAs are alcohols that could be used as fuel additives. Rate coefficients (in $cm^3$ molecule$^{-1}$ s$^{-1}$ unit) measured at ∼298K and atmospheric pressure (720 ± 20 Torr) were as follows: $k_1$ (E-4-methyl-cyclohexanol + Cl) = (3.70 ± 0.16) × $10^{-10}$, $k_2$ (E-4-methyl-cyclohexanol + OH) = (1.87 ± 0.14) × $10^{-11}$, $k_3$ (E-4-methyl-cyclohexanol + NO$_3$) = (2.69 ± 0.37) × $10^{-15}$, $k_4$ (3,3-dimethyl-1-butanol + Cl) = (2.69 ± 0.16) × $10^{-10}$, $k_5$ (3,3-dimethyl-1-butanol + OH) = (5.33 ± 0.16) × $10^{-12}$, $k_6$ (3,3-dimethyl-2-butanol + Cl) = (1.21 ± 0.07) × $10^{-10}$ and $k_7$ (3,3-dimethyl-2-butanol + OH) = (10.50 ± 0.25) × $10^{-12}$.  
[revised manuscript text omitted]

**S2. Structure-Activity Relationship (SAR) method**

SAR method allows to estimate a rate coefficient of an organic compound from its structure. The only possibility of the reaction of the studied compounds in this work with the atmospheric oxidants is the abstraction of an hydrogen atom. Consequently, the estimated rate coefficients of SAs are obtained from the sum of the rate coefficients for the H-atom abstraction from the primary ($k_{prim}$ (CH$_3$- )), secondary ($k_{sec,}$ (-CH$_2$-)) and tertiary ($k_{tert}$ (-CH<)) groups and from the alcohol ($k_{OH}$ (-OH)) group, taking into account the influence of the substituents attached to these groups, through substituent factors F(X), F(Y) and F(Z) (Equation S4).

$$k_{abs} = \sum k_{prim}F(X) + \sum k_{sec}F(X)F(Y) + \sum k_{tert}F(X)F(Y)F(Z) + \sum k_{OH} \qquad (S4)$$

At 298K rate coefficients for H-atom abstraction (in units of cm$^3$molecule$^{-1}$s$^{-1}$) and the reactivity factor for the reaction with OH are $k_{prim} = 1.36 \times 10^{-13}$; $k_{sec} = 9.34 \times 10^{-13}$; $k_{tert} = 1.94 \times 10^{-12}$ and $k_{OH} = 1.4 \times 10^{-13}$; F(CH$_3$) = 1; F(-CH$_2$-) = F(-CH<) = F(>C<) = 1.23 and F(-OH) = 3.5 from AOPWIN. The parameters for the reaction with Cl atoms are $k_{prim} = 2.84 \times 10^{-11}$; $k_{sec} = 8.95 \times 10^{-11}$; $k_{tert} = 6.48 \times 10^{-11}$ (in units of cm$^3$molecule$^{-1}$s$^{-1}$); F(CH$_3$) = 1; F(-CH$_2$-) = F(-CH<) = 0.8 and F(-OH) = 1.18 from Calvert et al. 2011. By last, the parameters used for the reaction with NO$_3$ radicals are $k_{prim} = 1 \times 10^{-18}$; $k_{sec} = 2.56 \times 10^{-17}$; $k_{tert} = 1.05 \times 10^{-16}$ and $k_{OH} = 2 \times 10^{-17}$ (in units of cm$^3$molecule$^{-1}$s$^{-1}$); F(CH$_3$) = 1; F(-CH$_2$-) = 1.02; F(-CH<) = 1.61; F(>C<) = 2.03 and F(-OH)=18 from Kerdouci et al. 2010, 2014.

The calculations for 4-methyl-cyclohexanol are the following:

[Figure]

$k_1 = k_{tert} \times F(-OH) \times F(-CH_2-) \times F(-CH_2-)$

$k_2 = k_3 = k_5 = k_6 = k_{sec} \times F(-CH<) \times F(-CH_2-)$.

$k_4 = k_{tert} \times F(-CH_3) \times F(-CH_2-) \times F(-CH_2-)$

$k_7 = k_{prim} \times F(-CH<)$

$k_{4MCHexOH+Cl} = 3.42 \times 10^{-10}$ cm$^3$molecule$^{-1}$s$^{-1}$

$k_{4MCHexOH+OH} = 1.92 \times 10^{-11}$ cm$^3$molecule$^{-1}$s$^{-1}$

$k_{4MCHexOH+NO3} = 2.27 \times 10^{-15}$ cm$^3$molecule$^{-1}$s$^{-1}$

The calculations for 3,3-dimethyl-1-butanol are the following:

$k_1 = k_{sec} \times F(-OH) \times F(-CH_2-)$

$k_2 = k_{sec} \times F(>C<) \times F(-CH_2-)$

$k_3 = k_4 = k_5 = k_{prim} \times F(>C<)$

$k_{3,3DM1ButOH+Cl} = 2.10 \times 10^{-10}$ cm$^3$molecule$^{-1}$s$^{-1}$

$k_{3,3DM1ButOH\ OH} = 6.08 \times 10^{-12}$ cm$^3$molecule$^{-1}$s$^{-1}$

$k_{3,3DM1ButOH\ +NO3} = 0.55 \times 10^{-15}$ cm$^3$molecule$^{-1}$s$^{-1}$

The calculations for 3,3-dimethyl-2-butanol are the following:

$k_1= k_{prim} \times F(-CH<)$

$k_2=k_{tert} \times F(>C<) \times F(-CH_3) \times F(-OH)$

$k_3= k_4= k_5= k_{prim} \times F(>C<)$

$k_{3,3DM2ButOH+Cl}=1.52 \times 10^{-10}$ cm$^3$molecule$^{-1}$s$^{-1}$

$k_{3,3DM2ButOH\ OH}=9.16 \times 10^{-12}$ cm$^3$molecule$^{-1}$s$^{-1}$

$k_{3,3DM2ButOH\ +NO3}=3.86 \times 10^{-15}$ cm$^3$molecule$^{-1}$s$^{-1}$

**Tables**

[revised manuscript text omitted]

A

[Figure]

B

[Figure]

**Fig. S5. A)** Residual FTIR spectra obtained in the reaction of 3,3DM1ButOH with (a) Cl atoms , (b) Cl atoms in the presence of NO , (c) OH radicals and (d) NO$_3$ radicals . The IR absorption bands subtracted were:3,3DM1ButOH, HCl, ClNO$_2$, ClNO, HCOH, HCOOH, HONO, NO$_2$, NO, N$_2$O and peroxy nitric acid (for Cl in the absence and presence of NO); N$_2$O$_5$, HNO$_3$, NO$_2$ (for NO$_3$ reactions) and HCOH, HCOOH, HNO, NO$_2$, CH$_3$ONO and CH$_3$ONO$_2$ (for OH reactions). (e) FTIR reference spectrum of 3,3-dimethylbutanal. **B)** Residual FTIR spectra: (a) Cl atoms , (b) Cl atoms in the presence of NO. (g) OH radicals (g) and (h) NO$_3$ radicals without 3,3-dimethylbutanal. Reference spectra (c)  acetone  from a commercial sample; (d) 2-methylpropanal ; (e) PAN ; (f) glycolaldehyde ; and (i) isobutylnitrate  from Eurochamp 2020 database.

[Figure]

**Fig. S6.** SPME/GC-TOFMS chromatograms obtained for the reaction of of 3,3DM1ButOH with Cl atoms , Cl atoms in the presence of NO, OH radicals and NO₃ radicals (30 min) and reference chromatograms of 3,3DM1ButOH and 3,3-dimethylbutanaldehyde.

[Figure]

**Fig. S7.** EI-MS spectra of the peaks of chromatograms shown in Fig. S6 obtained for the reaction of of 3,3DM1ButOH with Cl atoms, Cl atoms in the presence of NO+ NO, HOH radicals and NO₃ radicals. (a) $t_R$= 6.00 min; (b) $t_R$= 8.61 min; (c) $t_R$= 13.17 min; (d) $t_R$= 5.05 min).

**Fig. S8**. Reaction mechanism for degradation of 3,3-dimethylbutanal with the atmospheric oxidants in presence of NOx. H-Atom abstraction from the -COH group in 3,3-dimethylbutanal. Aschamnn et al., 2010.

[Figure]

**Fig. S9**. A) FTIR spectra obtained in the reaction of 3,3DM2ButOH with (a) Cl atoms(a), (b) Cl atoms in the presence of +NO, (b), (c) OH radials and (d) (c) NO₃ radicals(d) at 5 minutes of reaction. (e) FTIR reference spectrum of 3,3-dimethyl-2-butanone. B) FTIR spectra obtained in the reaction of 3,3-dimethyl-2-butanol with (a) Cl atoms(a), (b) Cl atoms in the presence of NO + NO (b), 25 minutes and 35 minutes of reactions respectively. (c) IR PAN spectrum. C) Residual FTIR spectra after subtraction of all known bands. (a) Cl atoms (a), (b) Cl atoms in the presence of +NO (b), (c) HOH ( radicals c) and (d) NO₃ radicals(d).

[Figure]

**Fig. S10.** SPME/GC-TOFMS chromatograms obtained for the reaction of of 3,3DM2ButOH with Cl atoms, Cl atoms in the presence of NO, OH and NO₃ radicals and reference chromatograms of 3,3DM2ButOH and 3,3-dimethyl-2-butanone.

| $t_R$ (min) | EI MS |
|---|---|
| 2.16 |
[Figure]
 |
| 5.39 ? | 2,2-dimethylpropanal |
| 6.03 | 3,3-dimethyl-2-butanone |
| 6.22 | Acetic Acid (SI 80%) |
| 6.46 | SPME |
| 6.96 | Nitrated compound |
| 7.75 | |
| 8.37 | |

[Figure]

| 9.86 | MS[1];9.85..9.87;-1.0*MS[1];9.93..10.06; / EI+(eiFi) / 130618_fotolisis 2 |
| 10.34 | MS[1];10.29..10.39;-1.0*MS[1];10.51..10.88; / EI+(eiFi) / 130618_fotolisis 2 |
| 13.46 | MS[1];13.42..13.52;-1.0*MS[1];13.59..14.48; / EI+(eiFi) / 130618_fotolisis 2 |

**Fig. S11.** EI-MS spectra of the peaks of chromatograms shown in Fig. S10 obtained for the reaction of 3,3DM2ButOH with Cl atoms , Cl atoms in the presence of ‒NO, HOH and NO$_3$ radicals, .

A)

[Figure]

**(B)**

[Figure]

**Fig. S12.** Concentration-time profiles of 3,3DM2ButOH and reaction products formed for the reaction of 3,3DM2ButOH with Cl atoms in  absence (A) and  presence of NO (B).

**Fig. S13**. Reaction mechanism for degradation of 3,3-dimethyl-2-butanone with the atmospheric oxidants in presence of NOx.